# Consensus-building conversation leads to neural alignment

**Beau Sievers** [1,2] ✉, **Christopher Welker** [1], **Uri Hasson** [3], **Adam M. Kleinbaum** [4] **& Thalia Wheatley**[1,5]

Conversation is a primary means of social influence, but its effects on brain activity remain unknown. Previous work on conversation and social influence has emphasized public compliance, largely setting private beliefs aside. Here, we show that consensus-building conversation aligns future brain activity within groups, with alignment persisting through novel experiences participants did not discuss. Participants watched ambiguous movie clips during fMRI scanning, then conversed in groups with the goal of coming to a consensus about each clip's narrative. After conversation, participants' brains were scanned while viewing the clips again, along with novel clips from the same movies. Groups that reached consensus showed greater similarity of brain activity after conversation. Participants perceived as having high social status spoke more and signaled disbelief in others, and their groups had unequal turn-taking and lower neural alignment. By contrast, participants with central positions in their real-world social networks encouraged others to speak, facilitating greater group neural alignment. Socially central participants were also more likely to become neurally aligned to others in their groups.

We take for granted that conversation plays a profound role in shaping belief and coordinating behavior. We rely on consensus-building conversation when the stakes are high, as in jury trials, scientific funding decisions, corporate boardrooms, and elections. Conversation is also pervasive in day-to-day life, from bars and coffee shops to workplace meetings[1]. Public conversation is the basis of deliberative democracy[2], and disruption of public conversation is considered a serious political threat[3]. Yet, we lack scientific understanding of how conversation shapes our beliefs and brain responses over time. How does talking change our thinking?

Previous research has shown that conversation determines important attitudes and behaviors. Even brief conversations can have meaningful effects, such as creating support for anti-discrimination laws[4] or influencing voting behvior[5]. Further, these effects can spread through social networks. Talking to a friend about voting may affect not only who they vote for, but who their future conversation partners vote for[6,7]. Remarkably, even brief social contact can influence the

health outcomes of friends and friends of friends[8,9], including depression[10], and drug and alcohol use[11–14].

However, it remains unclear whether these conversations shape the way individuals fundamentally see the world, as behavior is not a perfect reflection of deeply held belief—private acceptance is distinct from public compliance[15]. Psychologists, economists, and others offering explanations of social influence have therefore often treated conversation as a means of achieving public compliance, setting private beliefs aside. This may be because public compliance is easier to measure.

Recent studies suggest that neural alignment, measured as inter-subject correlation (ISC) of blood-oxygen level dependent signal from functional magnetic resonance imaging (fMRI)[16], can be used as a proxy measure of private acceptance. Although we cannot look at any one person's brain and know the exact contents of their thoughts, we can measure the similarity of activity across brains. For example, Chen et al.[17] used ISC to show that the structure of memory was shared

[1]Department of Psychological and Brain Sciences, Dartmouth College, Hanover, NH 03755, USA. [2]Department of Psychology, Stanford University, Stanford, CA 94305, USA. [3]Princeton Neuroscience Institute and Department of Psychology, Princeton University, Princeton, NJ 08544, USA. [4]Tuck School of Business, Dartmouth College, Hanover, NH 03755, USA. [5]Santa Fe Institute, Santa Fe, NM 87501, USA. ✉e-mail: beau@beausievers.com

among those with similar experiences. ISC has also been used to draw out inter-individual differences. When some movie-viewing participants pretended to be detectives while others pretended to be interior decorators, the decorators' brains systematically differed from the detectives' brains[18]. Likewise, when two groups were given different interpretations of an ambiguous story, alignment of brain activity was higher among people given the same interpretation[19]. A similar pattern of results held when participants invented their own interpretations of movie depicting interacting abstract shapes[20].

In these prior studies, experimenters either dictated an interpretation to a group or individuals reached an interpretation in isolation, eliminating the natural, back-and-forth dynamic of mutual social influence. But studies of brain alignment need not be limited to top-down instruction in artificial settings. For example, brain alignment in the classroom has been associated with student engagement[21] and retention of course content[22]. Interestingly, Parkinson and colleagues[23] found that friends in a real-world social network were more likely to have similar brain activity, suggesting brain alignment may play a role in determining who befriends whom, perhaps in conjunction with social reward systems[24]. Previous research has also shown that personality traits[25–28] predict network centrality, suggesting that alignment of brain activity within social networks may depend on the social behaviors of individuals.

Here, we show that groups who have consensus-building conversations become more neurally aligned relative to groups who do not reach consensus and controls who do not converse. Drawing participants from real-world social networks allows us to identify relationships between conversation behavior, network centrality, and changes in neural alignment toward or away from individual participants. The anatomical locations of neural alignment differ across groups and conversations, but are concentrated in brain areas associated with visual attention and movement, narrative understanding, and memory. Conversation behavior is also associated with differences in neural alignment: Groups with participants perceived as having high social status show unequal turn-taking and lower alignment, whereas groups with participants who were more central in their real-world social networks show greater alignment. This may be because perceived high-status participants signal disbelief in others' proposals and speak more, disrupting group consensus. By contrast, high-centrality participants encourage others to speak and are more likely to become neurally aligned with their groups. We discuss the implications of these results for understanding how cognitive processes shape the structure of social networks, as well as for theories of social influence, language, and the mind.

## Results

### Experimental paradigm

In Session 1, participants ($n = 49$, 23 male, 26 female based on self-reported free response; age range 26–32, mean age = 27.66)) watched movie clips with ambiguous narratives during brain scanning using fMRI. Afterward, participants answered a survey assessing their beliefs about each clip's narrative. In Session 2, participants met in small groups (9 groups; mean group size;= 4.2) to discuss the movie clips with the goal of reaching a consensus. Group membership was randomly assigned, pursuant to participants' scheduling constraints. Each group answered the survey presented in Session 1, but expressing the shared view of the group. Participants then rated the influence of the other participants and indicated their personal level of agreement with the consensus. In Session 3, participants re-watched the movie clips during fMRI scanning, along with additional novel clips featuring the same characters. Participants then answered a survey assessing their beliefs about the novel clips. A control group ($n = 9$) skipped Session 2, doing both fMRI sessions without the intervening group conversation.

To facilitate testing our social network centrality hypothesis, all participants were Master of Business Administration students at a private university in the United States. Because the university was rural and relatively isolated, the students formed a tight-knit community and, as part of their coursework, answered a survey used to map their cohort's social network.

For a complete description of the experimental procedure, see Methods.

### Consensus-building conversation aligned future brain activity

All participants but two reported agreeing with their group's consensus (as rated from −3 to +3 with values > 0 indicating more agreement; 1-sample $t(28) = 8.32$, two-tailed parametric $p < 0.001$, mean = 1.71, 95% CI = [1.29, 2.13]). Further, each group converged on a different consensus: after conversation, participants' survey answers became more similar to the answers of their conversation group members, compared to those in other groups. Hierarchical linear regression was used to test the effect of session (before versus after) and comparison type (within versus between groups) on the city block distance between subjects' survey answers (behavioral distance). The model included both predictors and their interaction, as well as random intercepts for participant pairs. The model significantly explained variance in behavioral distance (marginal $R^2 = 0.28$, two-tailed permutation $p < 0.001$, $n = 1369$ participant pairs). Session, comparison type, and their interaction all significantly predicted behavioral distance. Distance was higher before conversation ($\beta = 2.78$, 95% CI = [2.48, 3.08], two-tailed permutation $p < 0.001$), lower within groups ($\beta = -3.67$, 95% CI = [−4.4, −2.94], two-tailed permutation $p < 0.001$), and higher within groups before conversation ($\beta = 3.67$, 95% CI = [2.67, 4.67], two-tailed permutation $p < 0.001$) (Supplementary Figure 1).

Increased inter-subject correlation of fMRI BOLD signal (ISC) was observed within conversation groups, supporting the hypothesis that consensus-building conversation can align future brain activity. To capture change in ISC that was convergent across groups, we tested the effect of being in any conversation group (Fig. 1 bottom left, Supplementary Figure 2). Importantly, this analysis could not show the effect of being in a specific group that conversed about a specific movie clip. To address this limitation, we tested the effects of discussing specific movies with specific groups, counting the number of groups with statistically significant results (Supplementary Fig. 4). We use the term "movie–group combination" to refer to effects assessed for a specific group watching a specific movie; 5 movies times 9 groups gives 45 possible movie–group combinations. A wider range of brain areas were significant at the movie–group combination level, indicating that the neural effects of conversation depended on who was speaking and what they were speaking about.

In conversation groups, alignment tended to increase in visual and auditory sensory areas, as well as in higher-order areas associated with the attention and default mode networks, including the temporal parietal junction, angular gyrus, posterior cingulate, medial prefrontal cortex, and temporal pole. These results stand in contrast to the control group (no conversation), where ISC mostly decreased (Supplementary Fig. 3). See Supplementary Material for brain maps, Neurosynth terms (described below), change in alignment over time, and visualization of group convergence for each brain area with statistically significant change in alignment.

Importantly, the effects of consensus-building conversation can generalize to new stimuli. When viewing previously unseen clips sampled from later in each ambiguous movie, neural alignment was significantly higher within conversation groups. Across groups, higher within-group alignment for novel movies was observed in bilateral superior frontal gyrus (Supplementary Fig. 5). At the movie–group combination level, reflecting alignment unique to each group, after-conversation alignment was observed in a wide network of brain regions (Supplementary Fig. 6), consistent with the hypothesis that conversation can affect many future cognitive processes. Multiple regression was used to localize brain areas where ISC during viewing of

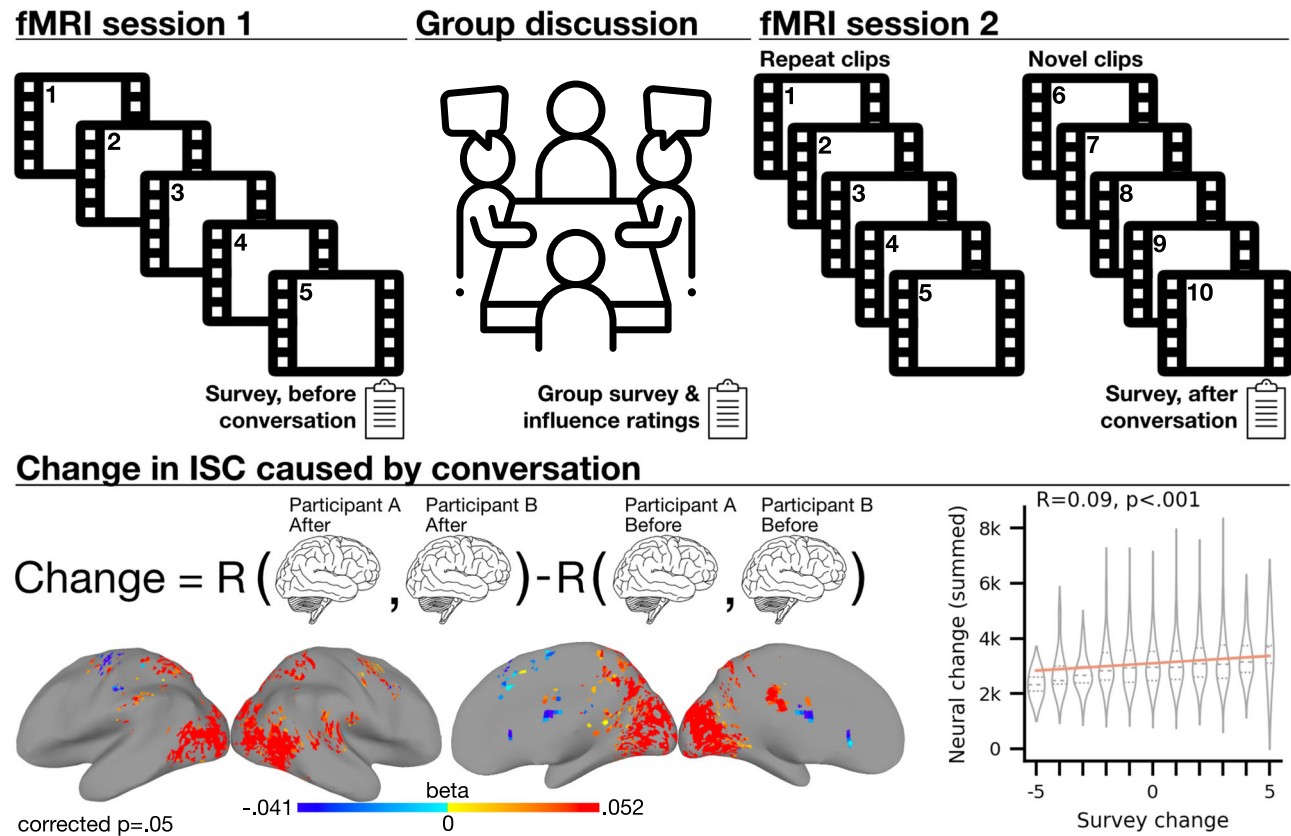

**Fig. 1 | Method and change in alignment across all movie clips and groups.** Top: Participants ($n = 49$) viewed five ambiguous movie clips during brain scanning, then met in small groups and discussed the movies with the goal of reaching a consensus interpretation. Participants re-watched the movie clips during brain scanning, as well as novel clips from later in each movie. At each step, participants filled out a survey capturing their interpretations. Linear regression was used to model change in inter-subject correlation (ISC) (see Methods). Bottom left: Change in neural alignment caused by consensus-building conversation. Color shows the multiple regression beta weight for being in any conversation group, across all five movie clips ($n = 703$ unordered participant pairs from 9 conversation groups, control group excluded). Results were thresholded at a two-tailed permutation test $P$-value of 0.05 corrected for multiple comparisons. Brain maps created using AFNI/SUMA.

Bottom right: Participants whose survey answers became similar showed greater neural alignment ($n = 3478$ unordered participant pairs across all movie clips, including control group participants that did not converse). The central diagonal line and shaded region show the regression line of best fit and its 95% confidence interval. Violin plots use width to represent the density of the distribution, with a central dashed line showing the median and dotted lines showing the lower and upper quartiles. "Brain" icon by Clockwise from Noun Project, available at https://thenounproject.com/icon/brain-1080481/. "Meeting" icon by SBTS from Noun Project, available at https://thenounproject.com/icon/meeting-5279011/. "Clipboard" by Made by Made from Noun Project, available at https://thenounproject.com/icon/clipboard-674066/. "Film" icon by NeueDeutsche from Noun Project, available at https://thenounproject.com/icon/film-531914/.

novel clips was predicted by conversation; for all conversation groups, and then for each movie–group combination (see Methods: Conversation-induced change in ISC).

Change in ISC was calculated by subtracting ISC before conversation from ISC after conversation, for each pair of participants. Multiple regression with multiple comparisons correction was used to localize change in ISC unique to each movie–group combination or present in all conversation groups. fMRI regression analyses were controlled by an intercept term corresponding to all within- and between-group participant pairs, including control group participants. This intercept term controlled for the effect of watching the movie clips twice, with or without conversation, regardless of conversation group. Unless otherwise noted, the reported results are relative to this intercept. During some movie clips, the control group showed some localized positive changes in ISC. This may be because some information was shared within most conversation groups, but not among controls, or because simply having a conversation, regardless of content, changed how participants engaged with the movie clips.

A permutation testing method that accounted for the grouped structure of the data was used to limit the false positive rate[29]. Because groups were small and conservative multiple comparisons correction was used, it is possible that this analysis did not detect all changes in

neural alignment. The scope of inferential statistical generalization is limited to the specific movie clips and groups reported here.

Similarity of survey answers was correlated with whole-brain alignment, even across groups, including control participants that did not converse ($R(3476) = 0.09$, two-tailed permutation $p < 0.001$, $n = 3478$ unordered participant pairs across all movie clips). To account for the non-independence of participant pairs, a mixed-effects model with random intercepts for participant pairs was used to estimate the effect of meaning change on whole-brain alignment, finding similar results (standardized $\beta = 0.1$, 95% CI = [0.07, 0.13], two-tailed parametric $p < 0.001$). These findings suggest that the timings and locations of neural alignment were not idiosyncratic, but were driven by convergence of beliefs (see Supplementary Material). Whole-brain alignment was calculated as the sum of unthresholded positive change in ISC. Note that while most statistically significant regions showed increases in alignment, some regions showed decreases.

See Supplementary Data 1 for a table of significant clusters across all fMRI analyses, corrected for multiple comparisons (cluster forming threshold: $p = 0.01$, minimum cluster size: 32 voxels, $p < 0.05$ corrected), with mean, peak, and 95% CI columns describing beta estimates for each significant cluster and atlas labels from Destrieux, Fischl, Dale, & Halgren[30].

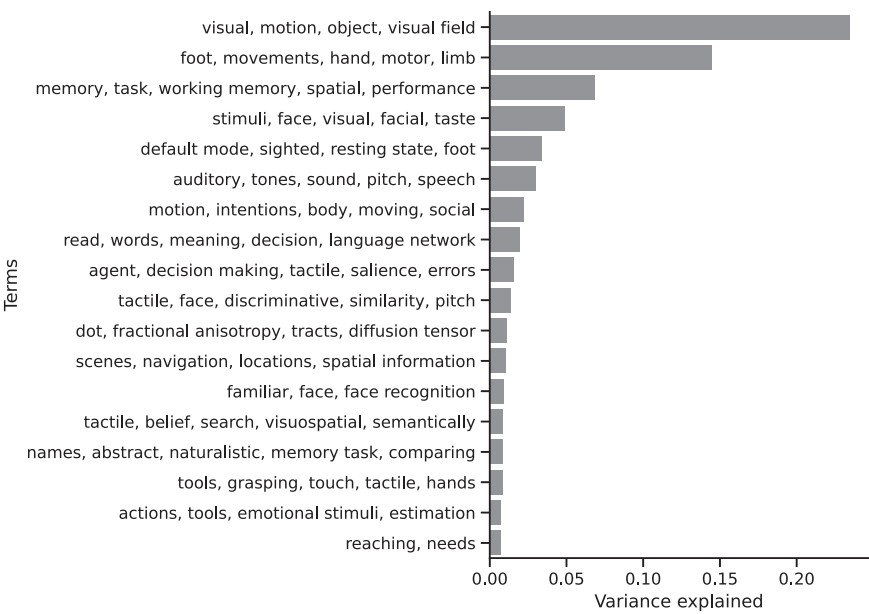

**Fig. 2 | Cognitive processes associated with aligned brain areas.** Conversation-related neural alignment was detected in brain areas associated with a range of cognitive processes, as identified by quantitative reverse inference using Neurosynth[31].

## Conversation aligned brain areas associated with a range of cognitive processes

After-conversation alignment was observed in brain areas associated with a wide range of cognitive processes, as estimated by quantitative reverse inference using Neurosynth[31]. Neurosynth uses a large database of previously published brain imaging literature, where English-language terms are associated with brain activation likelihood maps. For each statistically significant brain area in all reported fMRI analyses, we estimated which Neurosynth terms were a likely match. We then identified groups of terms that tended to co-occur using principal component analysis. 18 groups of terms explained 70% of the variance in the term probability data (Fig. 2). The group of terms that explained the most variance included words related to vision and motion, suggesting that one important function of conversation is future alignment of visual attention. Other high-ranking principal components were associated with motor activity, working memory, face perception, the default network, auditory perception, body movement and social understanding, and language. See Supplementary Material for term probabilities for each brain region and Methods for additional detail.

## Neural influence and social network centrality

The change analysis reported above could not capture directional social influence. Consider the following hypothetical example: Amar influences Beth, and later in the same conversation Carlos influences Amar. The method used above would detect that Carlos and Amar had aligned, but would overlook Amar's influence on Beth. To measure the degree to which each participant influenced each other participant we performed a neural influence analysis. Each participant was analyzed as both the person exerting influence (called the *ego*) and the person being influenced (called the *alter*), accounting for the possibility that participants could both influence others and be influenced by others in different respects. The amount of neural influence was the extent to which the alter became more similar to the ego's initial pattern of brain activity. Neural influence maps were calculated by comparing the brain activity of the ego before conversation to that of the alter both before and after conversation. Whole-brain neural influence was calculated as the sum of positive values in the unthresholded neural influence maps. Participants' centrality in the social network of their school cohort was measured using brokerage and eigenvector centrality. Brokers are

those who connect people who would not otherwise be connected, allowing them to control the flow of information between cliques[32]. People with high eigenvector centrality are both well-connected and have many well-connected friends[33]. Because of hypothesized differences between brokerage and eigenvector centrality[32,34], we tested the unique contributions of these measures. We also computed a *PCA centrality* score capturing variation common to both eigenvector centrality and brokerage (see Methods).

Unexpectedly, participants who were central in their social networks were more likely to be neurally influenced by others in their conversation groups (Fig. 3, Supplementary Figs. 7–10, Supplementary Table 8, Supplementary Data 1). Multiple regression was used to locate brain areas where neural influence was predicted by the PCA centrality, eigenvector centrality, and brokerage of egos and alters in the same conversation group, across all groups and movie clips (see Methods).

Ego PCA centrality predicted negative neural influence in right middle temporal gyrus (Supplementary Fig. 7), while alter PCA centrality predicted positive neural influence across a range of brain areas (Supplementary Fig. 8). Similarly, ego eigenvector centrality predicted negative neural influence in left middle temporal gyrus (Supplementary Fig. 9), while alter eigenvector centrality predicted positive neural influence across a range of brain areas (Supplementary Fig. 10). These results suggest variance in each centrality measure was associated with different neural processes. Analyses of neural influence specific to each group–movie combination yielded qualitatively similar results (Supplementary Figs. 11 and 12).

To detect neural influence that occurred in different parts of the brain for different pairs of participants, we tested whether network centrality predicted whole-brain neural influence. A mixed-effects model including eigenvector centrality and brokerage for both egos and alters as predictors modestly predicted whole-brain influence (marginal $R^2 = 0.03$, two-tailed permutation $p = 0.001$). Alter eigenvector centrality was the only significant predictor, and was associated with more whole-brain influence ($\beta = 0.43$, 95% CI = [0.12, 0.74], two-tailed permutation $p = 0.007$). No ego centrality measures significantly predicted whole-brain influence, and a separate mixed-effects model analysis showed that neither ego nor alter PCA centrality significantly predicted whole-brain influence. To account for the non-independence of participant pairs, these models included random

## Neural influence

**Fig. 3 | Neural influence across all movie clips and groups.** Changes in neural alignment toward or away from individual participants was captured by a neural influence measure. For each participant pair, one was designated the ego while the other was designated the alter. Neural influence reflects the movement of the alter's BOLD time series toward the ego's initial BOLD time series. When egos had higher PCA centrality, alters moved away from them after conversation (top brain map, blue areas). When alters had higher PCA centrality, they moved toward their egos after conversation (bottom brain map, red areas). All movie clips and groups were analyzed using a single regression model (see Methods). Brain maps ($n = 1406$ ordered participant pairs from 9 conversation groups, control group excluded) show the beta weights for ego PCA centrality (middle) and alter PCA centrality (bottom) (see Methods). Brain maps created using AFNI/SUMA. "Brain" icon by Clockwise from Noun Project, availble at https://thenounproject.com/icon/brain-1080481/.

intercepts for participant pairs. Participant pairs were included in these analyses only if they were in the same conversation group.

Suggesting that neural influence was related to changes in belief, survey influence calculated over the survey answers of ordered participant pairs was correlated with whole-brain neural influence ($R(8618) = 0.06$, two-tailed permutation $p < 0.001$, $n = 8620$ ordered participant pairs across all movie clips), and survey influence predicted neural influence in a mixed-effects model with random intercepts for participant pairs (standardized $\beta = 0.06$, 95% CI = [0.04, 0.08], two-tailed permutation $p < 0.001$) (see Supplementary Material for additional analyses comparing behavioral and neural results).

### Network centrality and perceived social status related to group alignment

Some groups showed stronger consensus and greater neural alignment than others. Exploratory hierarchical regression analysis showed that groups that spoke more had higher neural alignment ($\beta = 0.56$, 95% CI = [0.33, 0.79], two-tailed permutation $p < 0.001$). Groups with more high-centrality participants also showed higher neural alignment ($\beta = 0.52$, 95% CI = [0.3, 0.73], two-tailed permutation $p < 0.001$), and this effect was stronger the more words were spoken in conversation (2-way interaction, $\beta = 0.32$, 95% CI = [0.11, 0.53], two-tailed permutation $p = 0.04$). However, groups with unequal turn-taking had lower neural alignment ($\beta = -0.35$, 95% CI = [−0.57, −0.13], two-tailed permutation $p = 0.037$).

Groups with high-centrality participants had higher behavioral alignment ($\beta = 7.24$, 95% CI = [4.15, 10.33], two-tailed permutation $p < 0.001$). However, groups with high-centrality participants and also unequal perceived social status showed much lower behavioral alignment (interaction between centrality and Gini coef. of perceived social status, $\beta = -17.22$, 95% CI = [−24.77, −9.68], two-tailed permutation $p < 0.001$). To account for the hierarchical structure of the data, models included random intercepts for movie clips. Both regression models significantly explained variance in outcomes (neural

alignment: marginal $R^2 = 0.53$, two-tailed permutation $p < 0.001$, $n = 45$ group–movie pairs; behavioral alignment: marginal $R^2 = 0.34$, two-tailed permutation $p < 0.001$, $n = 45$ group–movie pairs).

Below, we describe similarities and differences between high PCA centrality and perceived high-status participants in their word choices and general conversation behaviors. For further description and examples of participants' conversation behavior, see Supplementary Material.

Word use in each speech turn was modestly predictive of participants' PCA centrality ($F(350, 7771) = 2.0$, two-tailed permutation $p < 0.001$, $R^2_{adj} = 0.04$) and perceived status ($F(350, 7771) = 2.39$, two-tailed permutation $p < 0.001$, $R^2_{adj} = 0.06$). For regression analysis details see Methods; see Supplementary Table 2 for significant word-level betas and P values and Supplementary Data 2 for a complete regression table.

To qualitatively understand how the identified words functioned in context, we searched the conversation transcripts and read the speech turns before, after, and containing each word. This revealed marked differences in how high-centrality and perceived high-status participants' behaved in conversation. High-centrality participants encouraged others to express themselves, whereas perceived high-status participants did the opposite.

While both high-centrality and perceived high-status participants used casual language like *cool, gotta, for sure, dude, shit, gonna*, and *'cause*, these words were stronger predictors of centrality than perceived status. High-centrality (but not perceived high-status) participants also spoke quite frankly, discussing *sexual* content ("There's gonna be so much tension it's gonna turn sexual"), describing ideas or movie clips as *weird* ("I had some really weird thoughts on this one"), and using *fuck* as an intensifier ("Weird fucking movie"). High-centrality participants also prompted others to comment on the group's survey answers using *far* ("So, what I have so far is…"), *answer* ("That's a good answer now"), and *character* (Reading a survey question aloud: "Are the characters discussing politics?"). These behaviors

may have made other participants more willing to express themselves. By contrast, perceived high-status (but not high-centrality) participants challenged others' proposals by asking *really?* (e.g.: Speaker 1: "Yeah, that's Joaquin Phoenix." Speaker 2: "Really?"). They also tended to express tentative or reserved approval of others' proposals using *fine* ("I can go with it. I'm fine."). These behaviors may have made others less willing to express themselves.

Pairs of hypothesis-blind coders classified each speech turn by its function in the conversation (see Methods). Perceived high-status participants were more likely to propose explanations of movie content ($\beta = -0.81$, 95% CI = [−1.17, −0.44], two-tailed parametric $p = 0.001$), more likely to direct attention to others ($\beta = 0.09$, 95% CI = [0.04, 0.13], two-tailed parametric $p < 0.001$), and were also more likely to give orders to others ($\beta = 0.1$, 95% CI = [0.08, 0.12], two-tailed parametric $p < 0.001$). Perceived high-status participants were also more likely to reject others' proposed explanations, favoring implicit rejection (e.g., by asking "Really?" or saying something that when evaluated in context suggests another's proposal is unlikely) ($\beta = 0.04$, 95% CI = [0.01, 0.07], two-tailed parametric $p = 0.016$). Perceived high-status participants were more likely to speak with confidence ($\beta = 0.02$, 95% CI = [0.02, 0.03], two-tailed parametric $p < 0.001$) and more likely to make jokes ($\beta = 0.01$, 95% CI = [0.0, 0.02], two-tailed parametric $p = 0.018$). Accordingly, speech turns by perceived high-status participants were slightly less likely to be ignored by others ($\beta = -0.01$, 95% CI = [−0.01, −0.01], two-tailed parametric $p < 0.001$).

Like perceived high-status participants, high-centrality participants were more likely to speak with confidence ($\beta = 0.02$, 95% CI = [0.0, 0.03], two-tailed parametric $p = 0.043$). However, high-centrality participants were more likely to ask others for clarification ($\beta = 0.16$, 95% CI = [0.06, 0.25], two-tailed parametric $p = 0.001$), were less likely to suggest changes to other speakers' proposals ($\beta = -0.13$, 95% CI = [−0.21, −0.05], two-tailed parametric $p = 0.003$), and were slightly less likely to have their proposals incorporated into the group explanation of the movie content ($\beta = -0.01$, 95% CI = [−0.02, −0.0], two-tailed parametric $p = 0.045$).

Groups with higher median perceived status had more unequal turn-taking ($R(43) = 0.35$, two-tailed permutation $p = 0.02$). Accordingly, percived high-status participants spoke more ($R(37) = 0.65$, two-tailed permutation $p < 0.001$) and were rated as more influential by others in their group ($R(27) = 0.74$, two-tailed permutation $p < 0.001$). Participants who spoke more were also rated more influential ($R(27) = 0.77$, two-tailed permutation $p < 0.001$). However, perceived status was negatively correlated with pairwise, whole-brain neural influence ($R(7028) = -0.05$, two-tailed permutation $p < 0.001$, $n = 7030$ unordered participant pairs across all movie clips) and a mixed-effects model trained with random intercepts for participant pairs yielded a similar result (standardized $\beta = -0.05$, 95% CI = [−0.085, −0.016], two-tailed parametric $p = 0.005$). The correlation between centrality and words spoken was not significantly different from zero ($R(37) = 0.24$, two-tailed permutation $p = 0.143$), and the correlation between centrality and influence ratings was not significantly different from zero ($R(27) = 0.15$, two-tailed permutation $p = 0.433$).

Taken together, these results suggest that the conversation behaviors of high-centrality participants supported group alignment, while the behaviors of participants with perceived high social status produced public compliance without private acceptance, and that this was detrimental to group alignment.

## Discussion

The naturalistic study design, in which participants showed higher neural alignment after discussing movies they just watched, exposes the power of conversation to shape how our brains process the world. Differences between conversation groups and controls show that consensus-building conversation strengthens the neural alignment of group members across a range of brain areas. These areas were associated with many cognitive processes, ranging from vision and audition to attention, language, and memory. Each group aligned in its own way, and group alignment was correlated with the degree of behavioral consensus. Further, increases in group alignment generalized to novel stimuli not discussed, indicating that conversation provided a mental framework to interpret new information. Importantly, though the present results show that consensus-building conversation can create neural alignment, the mechanistic details of this process remain unknown.

We did not expect all conversations to be equally successful, and some conversation groups aligned more than others. Specifically, groups with participants perceived as having high social status showed less alignment, while groups with high-centrality participants showed more alignment. This may be because perceived high-status and high-centrality participants behaved differently in conversation. Groups with higher median perceived status had more unequal turn-taking. This may be because perceived high-status participants spoke more, gave more orders, and implicitly rejected others' proposals. Perceived status was associated with higher influence ratings from group members, but lower neural influence measurements, raising the possibility that perceived high-status conversation behaviors produced public compliance without private acceptance. This misperception of status cues as markers of influence may play a pernicious role in the reinforcement of power hierarchies[35].

By contrast, real-world social network centrality was associated with neural influence measurements in both directions: high-centrality participants became more similar to their group members, while at the same time their group members became more similar to them. Accordingly, groups with high-centrality participants achieved greater neural alignment. High-centrality participants may have facilitated this alignment by creating a psychologically safe environment, encouraging others to speak[36], privately accepting and internalizing others' proposals, and rallying their groups around agreeable consensus positions.

These results suggest the possibility that thinking like one's conversation partners facilitates social connection. Previous research on personality and social network centrality points in this direction: People with high self-monitoring personalities (i.e., those who adapt their behavior to the people around them) tend to be more socially central[37,38], and they become so by making friends across disconnected cliques[28]. Further, survey measures of self-monitoring, cognitive flexibility, and communication flexibility are closely related and highly correlated[39,40]. Although we do not know how our participants became central in their social networks, it is plausible that the ability to help groups reach consensus through a combination of influence and flexibility enabled them to grow large and diverse groups of friends. It is also possible that those in central network positions for independent reasons are motivated to develop consensus-building conversation behaviors. By connecting the dots between how people think, how people speak, and who they socialize with, we hope to provide a cognitive, process-level view of social network structure.

As across the sciences, arts, and humanities, knowledge is increasingly produced by collaborative, conversing teams[41], these effects have practical implications for leadership and management, as well as for predicting the spread of information through social networks. The present results also fit research showing that powerful (i.e., socially central) individuals can afford to manage the conflicts of others, resulting in a benefit to their group[42,43].

### Limitations on generalization

Although the present results are suggestive, they are limited in scope. The reported effects are relatively small, and because movie clips and groups were treated as fixed effects, the statistical methods used do not guarantee generalization. Different groups discussing different movies may exhibit different patterns of

alignment and influence. In particular, we note that participants in the present study had very little moral stake in the movie clips discussed. Morally charged conversations may yield different results. Further, participants were given the explicit goal of coming to a consensus. While many natural conversations share this feature explicitly (e.g., jury deliberations) or implicitly (e.g., meetings at work), many other important conversation contexts are naturally adversarial (e.g., political debates), and in these contexts we should also expect different results. In particular, responsiveness to influence may be less desirable in contexts where groups seek a correct solution to a well-defined problem, or where there is direct competition within or between groups. Further, our participants were all Master of Business Administration students at a private university in the United States, and likely differed from the general population in socioeconomic status, general intelligence, and conversation behavior, limiting the generalizability of the reported findings. Finally, the conversation content analysis used a rubric to classify speech turns. We found that percieved high-status participants made more statements not captured by the rubric ($\beta = 0.02$, parametric $p = 0.007$), suggesting the rubric was not a complete catalog of relevant conversation behaviors. Uncovering generalizable factors affecting consensus and neural alignment will require future research.

### Theoretical implications: influence, language, and the mind

In agreement with prior studies[44,45], we found that the cognitive processes underlying social understanding were shared across people. Further, we found that these processes were more aligned after natural consensus-building conversation, not only when participants were listening to each other, but into the future. These results bear on theories of social influence, language and the mind.

Much research on social influence has focused on public compliance, setting aside the long-lasting effects of social interaction on private cognition[15,46–50]. However, the present results show that consensus-building conversation can align neural responses within groups, and that this alignment can generalize to novel stimuli that were not discussed. This suggests a stronger role for private acceptance in understanding social influence, and demonstrates the feasibility of fMRI as a tool for measuring changes in private thought[51].

The observed results could only have occurred if participants' ways of speaking were commensurable, complicating strong versions of accounts on which people regularly use the same words but with completely different meanings[52–54]. Additionally, we would not have observed neural alignment if the participants tended to represent the same concepts using different neural processes[55,56].

The reported results suggest the possibility that one function of consensus-building conversation is producing neural alignment. This would pose a challenge to the claim that language did not evolve for communication, but instead for organizing individual thought[57]. By contrast, the reported results are consistent with the theory that the coordination of belief is an important evolutionary function of language[58–64], and further suggest the alignment of neural processes governing attention as a mechanism. On this account, conversation resembles a neuro-feedback process[65] where people use language to understand and influence others' mental states. The tight, ever-evolving coupling between your thoughts and my thoughts, corresponding to tight neural alignment across our brains, is a plausible mechanism for building group realities based on shared language. This view is compatible with predictive processing accounts of cognition[66,67], especially those that place special importance on the challenge of predicting other people[68,69]. The aligning function of conversation may also support cumulative development on an evolutionary time scale, as the language environment created by past generations scaffolds the learning of the next[70–74].

## Methods

### Participants

Participants who completed the social network survey were first-year Master of Business Administration students at a private university in the United States ($n = 865$). The survey was completed as a part of their coursework on leadership. A subset of these participants ($n = 49$, 23 male, 26 female based on free-response self-report; age range 26–32, mean age = 27.66) went on to participate in the fMRI studies. We had no hypotheses concerning sex or gender, so sex and gender were not considered in the study design beyond recruiting a gender-balanced participant sample. Scheduling was facilitated by a custom web application that allowed participants to select available sessions. Participants met in small groups with 3–6 participants per group (9 groups; mean group size = 4.2). 9 participants were assigned to a separate control group that did not complete the conversation task. Participants were compensated with a cash payment and an animated digital image of their brain anatomy. All participants provided written informed consent, and all experimental procedures were reviewed and approved by the Dartmouth College Committee for the Protection of Human Subjects.

### Exclusions and missing data

A total of 59 participants underwent fMRI scanning, however, 10 participants were excluded, yielding 49 participants. 5 participants were excluded because of technical difficulties during scanning, 1 was excluded because the scanner compatible glasses were insufficient and they couldn't see actors' facial expressions, 2 were excluded because they terminated the scan session due to discomfort, 1 was excluded because they were absent from the group discussion, and 1 was excluded because an anatomical anomaly was detected (this participant was referred to a neurologist for follow-up in accordance with Dartmouth Brain Imaging Center safety policies). Due to a technical error, three groups were given an incorrect version of the survey for fMRI session 2 (after conversation) that did not include run-by-run agreement ratings or yes-or-no questions for the repeated movie clips. These same three groups were also not given pen-and-paper surveys after the group session, but instead verbally confirmed that they agreed with the group consensus (the realization that this was inadequate led to the introduction of the pen-and-paper survey).

### Session 1: before-conversation movie viewing

Participants viewed a selection of movie clips during fMRI scanning. Naturalistic, narrative movie clips were selected to capture the audience's attention over an extended period of time, maximizing the anatomical extent of neural alignment[75]. All movie clips focused on social interaction and were selected from a range of major motion pictures (see Movies, below). Importantly, all movie clips were narratively ambiguous, allowing participants to form a range of interpretations that could be plausibly be changed via social influence. In addition, the sound was turned off, removing music, dialog, and context cues that might otherwise constrain interpretation of the narrative. Two additional movie clips with interesting social content were selected for use in hyperalignment[76,77]. Hyperalignment clips were presented with the sound on. See Supplementary Table 7 for titles, presentation order, and duration in seconds and TRs. All movie clips were edited for time and narrative continuity. The edited clips may be downloaded at https://osf.io/kr9fb/.

The before-conversation fMRI session consisted of an anatomical scan, five echo-planar imaging (EPI) runs of ambiguous movie clip viewing, and two EPI runs of hyperalignment movie clip viewing (naturalistic movies with rich social interaction, with the sound turned on). After scanning, participants answered a detailed survey to assess their individual, before-conversation understanding of the movie clip content. Questions about each movie clip were accompanied by an array of screenshots to act as a memory aid.

### Session 2: group conversation

Participants met in groups of three to six and were seated around a circular table. The only instructions were to talk with the goal of reaching a group consensus, and to fill out the survey so it reflected that consensus. The survey included an array of screenshots for each clip as a memory aid and was presented on a laptop that could be moved around from participant to participant. Participants were allotted 15 minutes of conversation per movie clip and were given verbal warnings when the remaining time reached five minutes and one minute. If participants finished before the time limit was reached they informed the experimenter and moved directly on to discussing the next clip. After participants were finished, they each went into separate rooms and filled out pen-and-paper surveys rating their agreement with the group consensus and evaluating the influence of each group member (including themselves) on the group consensus.

### Session 3: after-conversation movie viewing

In the final fMRI scan session, participants viewed the same selection of clips presented during Session 1. Participants then viewed five more movie clips, each of which was a scene from later in each of the movies they had already seen. These clips were used to test the generalization of the group consensus to novel stimuli. After leaving the scanner, participants answered a survey containing the same questions as in Sessions 1 and 2, except for the new set of clips the participants had just seen.

### Movies

Gower, L., Morris, N., Piel, J-L. (Producers), & Glazer, J. (Director). (2004). *Birth*. United States: New Line Cinema.

Pitt, B., Gardner, D., Scott, R., Daly, J., Valdes, D. (Producers), & Dominik, A. (Director). (2007). *The Assassination of Jesse James by the Coward Robert Ford*. United States: Warner Bros. Pictures.

Thomas, J. (Producer), & Glazer, J. (Director). (2000). *Sexy Beast*. United Kingdom, Spain: Fox Searchlight Pictures.

Sellar, J., Lupi, D., Anderson, P. T., Ellison, M. (Producers), & Anderson, P. T. (Director). (2012). *The Master*. United States: The Weinstein Compay.

Cuarón, A., Vergara, J. (Producers), & Cuarón, A. (Director). (2001). *Y tu mamá también*. Mexico: 20th Century Fox, IFC Films.

**Hyperalignment movies.** Bialic, G. (Producer), Blichfeld, K., & Sinclair, B. (Directors). (2016). "High Maintenance" *Tick*. United States: Home Box Office.

Pollack, S., Samuels, S., Fox, J., Orent, K. (Producers), & Gilroy, T. (Director). (2007). *Michael Clayton*. United States: Warner Bros. Pictures.

### fMRI image acquisition

Participants were scanned at the Dartmouth Brain Imaging Center using a 3T Siemens Prisma scanner with a 32-channel head coil. A high resolution T1-weighted MPRAGE anatomical scan (2.32 ms TE; 2300 ms TR; 0.9 x;0.938 x 0.938 mm resolution) was performed at the beginning of each scanning session. Functional images were acquired using an echo-planar sequence (32 ms TE; 727 ms TR; 53° flip angle; 3 x 3 x 3 mm resolution). The number of scans per run varied depending on the stimulus presented. Sound was delivered using an in-ear headphone system. Foam padding was placed around participants' heads to minimize motion.

### fMRI image preprocessing

Anatomical images were deobliqued using AFNI `3dWarp`. Brain extraction was performed using ANTs `antsBrainExtraction.sh` with priors derived from the MICCAI 2012 Multi-Atlas Challenge Data [ref. 78; Available at: https://my.vanderbilt.edu/masi/workshops/]. Tissue segmentation was performed using FSL `fast`, and tissue masks

were saved for calculating tissue-specific nuisance variables (below). Anatomical images were normalized to the non-linear, asymmetrical MNI ICBM152 template[79] using ANTs `antsRegistrationSyN.sh`, and transformation matrices were saved for normalizing EPI time series (below).

EPI images were motion corrected using FSL `mcflirt` and motion outliers (framewise displacement > 0.9) were detected using FSL `fsl_motion_outliers`. Motion parameters and outlier TR indices were saved. EPI-to-anatomical transformations were calculated using AFNI `align_epi_anat.py` and saved for normalizing EPI time series (below). EPI images were deobliqued using AFNI `3dWarp`, then percentage-scaled[80] using NumPy, then normalized to the non-linear asymmetrical MNI ICBM-152 template by concatenating the EPI-to-anatomical and anatomical-to-MNI transformations and applying them in a single step using ANTs `antsApplyTransforms`. EPI images in MNI space were then iteratively blurred until reaching a smoothness of 6mm full width at half maximum using AFNI `3dBlurToFWHM`. Nuisance variables were removed from the smooth EPI images in MNI space using Nilearn `clean_img()`. Nuisance regressors included: 6 motion parameters, framewise displacement outliers (one binary regressor per outlier), tissue confounds[81], linear and quadratic trends, and an intercept term.

### Hyperalignment

Searchlight hyperalignment[82] was performed using PyMVPA[83]. Hyperalignment maps were calculated using runs 6 and 7 from the pre-conversation fMRI session. Because 6 participants did not complete both runs, a second set of hyperalignment maps was calculated using only run 6, which was completed by all participants. For all analyses, EPI images hyperaligned using both runs 6 and 7 were used if they existed; if not, the EPI image hyperaligned using only run 6 was used. To work around an issue in PyMVPA (described at: https://github.com/PyMVPA/PyMVPA/issues/589), very low-amplitude Gaussian noise was added to invariant features (voxels with no variance) before calculating the hyperalignment maps, and also before forward-mapping of EPI images. Inspection of the images showed that the invariant features were voxels included in the MNI ICBM-152 template mask, but not in individual participants' EPI images, because sometimes parts of the cerebellum were not imaged.

### Inter-subject correlation

The basic unit of the reported neuroimaging analyses is inter-subject correlation (ISC). Here, ISC is defined as the Pearson correlation of time series in corresponding voxels in a pair of participants. Voxel-wise ISC was calculated on the hyperaligned images using AFNI `3dTcorrelate`.

### Conversation-induced change in ISC

ISC was calculated for every participant pair, creating an ISC matrix. A *change* matrix was calculated by subtracting before-conversation ISC from after-conversation ISC. The effect of group membership on change in ISC was quantified using regression analysis. For the group-specific analyses, predictor matrices were created for each group, with 1s where both participants were in the target group and 0s otherwise. For the across-groups analysis, a single prediction matrix included 1s when both participants were in the same conversation group. For all analyses, an all-1s intercept matrix was used to capture change that occurred whether or not participants were in the same group (i.e., change caused simply by watching a movie clip twice). For analyses including all movie clips, the relevant change and predictor matrices were concatenated, supporting the analysis of multiple clips using a single model. Unraveled vector versions of these matrices were used as predictors in a multiple regression with the unraveled change matrix as the target, executed using NumPy `linalg.lstsq()`. The beta values from this regression represent the change in ISC either unique to each

## Calculating change in ISC

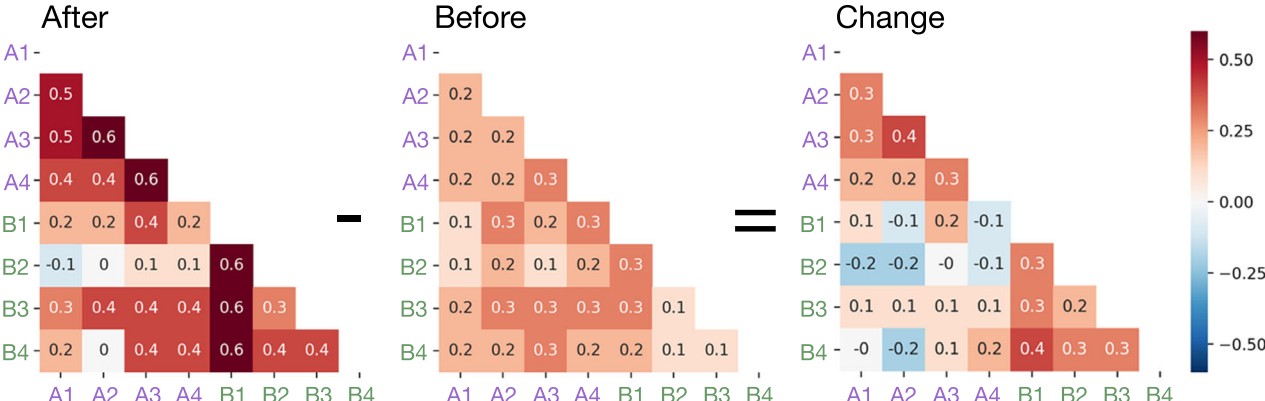

## Regression analysis

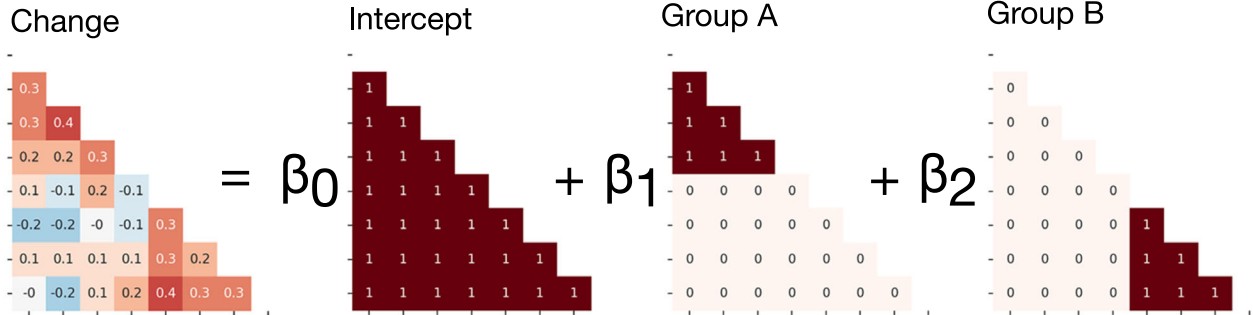

## Subject-wise permutation testing

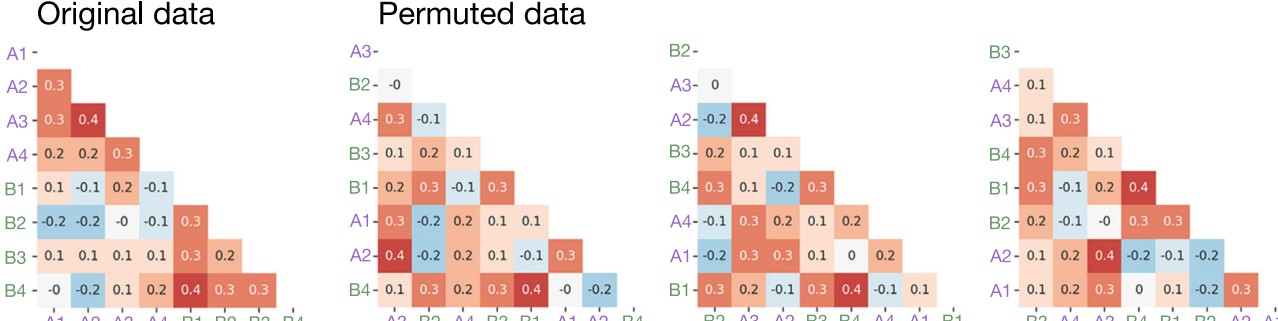

**Fig. 4 | Change analysis example.** A worked toy example of the change analysis for two groups each containing four participants. Top: ISC matrices were calculated for both fMRI sessions. Each matrix contained ISC measurements for all participant pairs. The before-conversation ISC matrix was subtracted from the after-conversation ISC matrix to obtain the *change matrix*. Middle: Structure of the multiple regression analysis. Predictor matrices captured change caused simply by watching movie clips twice (the intercept), as well as change caused by consensus-building conversation within a single group. Bottom: Examples of subject-wise permutation[29], where rows and columns are identically shuffled.

group or common across all conversing groups. See Fig. 4 for a worked example. Repeating this procedure for each movie clip across every voxel in the fMRI data yielded *change maps* showing where in the brain change occurred for each movie–group combination. See Results and Fig. 1.

To test whether conversation-related change extended to the novel stimuli shown in the after-conversation fMRI session, a

regression was performed using the same predictors, but with the after-conversation ISC matrix as the target.

Significance values were calculated using subject-wise permutation testing[29]. The regression described above was repeated 2000 times. Each time, the rows and columns of the change matrix were identically permuted (Fig. 4, bottom). This procedure satisfies the exchangability assumption of permutation testing, appropriately

limiting the false positive rate. Two-tailed *P*-values for each parameter were obtained by calculating the proportion of permutations where the true result was more extreme than the permuted results. Statistical significance was achieved when the true result was more extreme than 97.5% of the permuted results, establishing a two-tailed alpha value of 0.05. This approach controls the false positive rate given the pairs-and-groups structure of the study, but it is not equivalent to the use of a multilevel model. Multiple comparisons correction was performed at the cluster level using AFNI `3dClustSim`. Cluster simulation used a non-Gaussian ACF model allowing for heavy tails, using model parameters estimated from the mean of the collected fMRI data using AFNI `3dFWHMx`. Analyses including all groups and movies used a cluster-forming threshold of $p = 0.05$ and a minimum cluster size of 115, giving $p = 0.05$ corrected. Analyses of specific groups watching specific movie clips used a cluster-forming threshold of $p = 0.01$ and a minimum cluster size of 32, also giving $p = 0.05$ corrected.

To identify time points when change occurred, ISC was calculated in a rolling window for each significant cluster in both the before- and after-conversation fMRI sessions. The rolling window ranged 10 TRs and had a step size of 1 TR. The mean pairwise ISC calculated within each window was mapped to the TR corresponding to the window's center, creating a group-level ISC time series. The difference between the before-conversation group ISC time series and the after-conversation group ISC time series was used to calculate a group change time series. Group change time series for each significant cluster are included in Supplementary Material.

### Social network centrality

Exhaustive maps of participants' social networks were created using an online name generator survey. Participants in this study came from three separate cohorts of MBA students, and separate social networks were generated for each cohort. All first-year students in each cohort (Ns = 285, 293, 287; Fig. 5) were emailed a link to the survey website. The sole survey question was adapted from Burt[32] and read as follows: "Consider the people with whom you like to spend your free time. Since you arrived at [institution name], who are the classmates you have been with most often for informal social activities, such as going out to lunch, dinner, drinks, films, visiting one another's homes, and so on?" To reduce the likelihood of inadequate or biased recall, all possible classmates names were listed alphabetically in four columns, which each column corresponding to one of the MBA program's assigned sections. Participants checked a box next to a name to indicate the presence of a social tie. No limit was placed on the number of social ties or the time taken to complete the survey.

Survey responses were used to create directed graph representations of each network, where nodes corresponded to respondents and edges to social ties. Each cohort had a similar number of social ties/edges (9472, 7340, 7676) and diameters (3, 3, 2). Two metrics were computed over these graphs: eigenvector centrality, a measure of how well each node in a graph is connected to other well-connected nodes[33], and brokerage, a measure of how well each node connects other nodes that would not otherwise be connected[32]. Both measures were computed using the R package igraph[84]. Brokerage was calculated by raising the network's constraint to an exponent of −0.5. Because eigenvector centrality and brokerage are often highly correlated, principal component analysis was used create a centrality measure that captured the variance shared by both metrics. PCA was applied using Scikit-learn[85], and the social network metrics were projected on the first principal component. The resulting projection is referred to as *PCA centrality*. Participants in the fMRI study had widely varying centrality values, covering most of the population distribution (Fig. 5).

### Neural influence

We refer to the participant hypothesized to exert neural influence as the *ego*, and the participant hypothesized to be influenced as the *alter*.

First, ISC matrices including every possible ego–alter pair were calculated. The *initial position matrix* measured the ISC of the ego and the alter during the before-conversation fMRI session, while the *final position matrix* measured the ISC of the ego before conversation with the alter after conversation. To obtain an *influence matrix*, the initial position matrix was subtracted from the final position matrix (Fig. 6). Unlike the change analysis above, in which only the lower triangle of each matrix was used, in the influence analysis both the upper and lower triangles were used, to test both members of each pair in the ego *and* the alter position. This procedure yielded neural influence maps at the participant-pair level, with values directly interpretable as how much more similar the alter became to the ego's initial time series.

The effect of social network centrality was assessed using regression analysis over the neural influence maps generated at the level of the participant pair. Predictor matrices were structured similarly to the predictors used in the across-groups change analysis, but instead of 1s, each cell included the social network centrality values of either the ego or the alter. Group-specific predictor matrices zeroed out cells for all but the target group. For analyses including all movie clips, the relevant influence and predictor matrices were concatenated, supporting the analysis of multiple clips using a single model. Because predictor beta estimates made with ordinary least squares regression depend only on the variance unique to that predictor, this approach allowed us to estimate the effects of PCA centrality with one model, and then to estimate the unique effects of eigenvector centrality and brokerage with a second model. Each regression yielded maps of centrality-dependent neural influence.

The correlation of positive cluster values in the ego neural influence map with alter ratings of influence during the group conversation for each participant pair was used to test whether neural influence corresponded with social influence.

### Reverse inference with Neurosynth

For each significant region in all of the foregoing analyses and each of the 3228 Neurosynth terms, we obtained a vector of probabilities that a term would be used, given the spatial pattern of the result (i.e., the posterior probability(feature|activation), assuming an empirical prior). This yielded a matrix where rows corresponded to brain regions and columns to term posterior probabilities. Anatomical and duplicate terms were removed. The columns of the term probability matrix were reduced to 18 components using Principal Component Analysis, capturing 70% of the variance in the data.

### Conversation content analysis

All conversations were transcribed. Pairs of hypothesis-blind coders classified each speech turn by its function in the conversation. For a complete list of speech turn types, see Supplementary Table 4. Coders also rated each speech turn from 1 to 10 on a range of continuous properties. For a complete list of continuously rated speech turn properties, see Supplementary Table 6. Coders also rated participants' perceived social status, whether a speech turn referred to other speakers, whether a speech turn was prompted by another speaker, whether the speaker interrupted someone, and whether others laughed. Speech turn coding had moderate inter-coder reliability (median Cohen's kappa = 0.53, all parametric *P*-values < 0.001). Ten additional coders rated each participant's perceived social status. Perceived status coding had excellent inter-coder reliability (F(38, parametric $p < 0.001$, 570) = 10.72, ICC2k = 0.9, 95% CI = [0.85, 0.94]). Cohen's kappa for speech turn coding was computed using the `irr` package for R version 3.6.0 (2019-04-26), and intra-class correlation for status coding was computed using the `psych` package for R, based on a mean-rating, absolute-agreement, 2-way mixed-effects model. Similarly, participants' perceived status values were estimated using beta values from a mixed-effects model with z-scored status ratings as

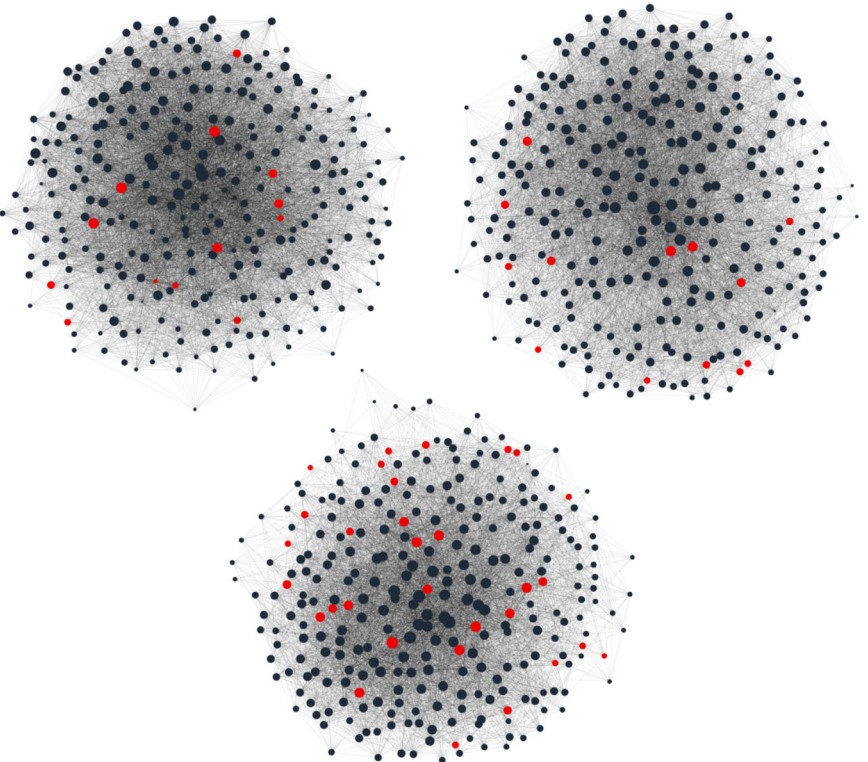

**Fig. 5 | Participant social networks.** Visualizations of the participants' social networks. Dot size is scaled by PCA centrality (see Methods). Red dots were participants.

## Calculating neural influence from ISC

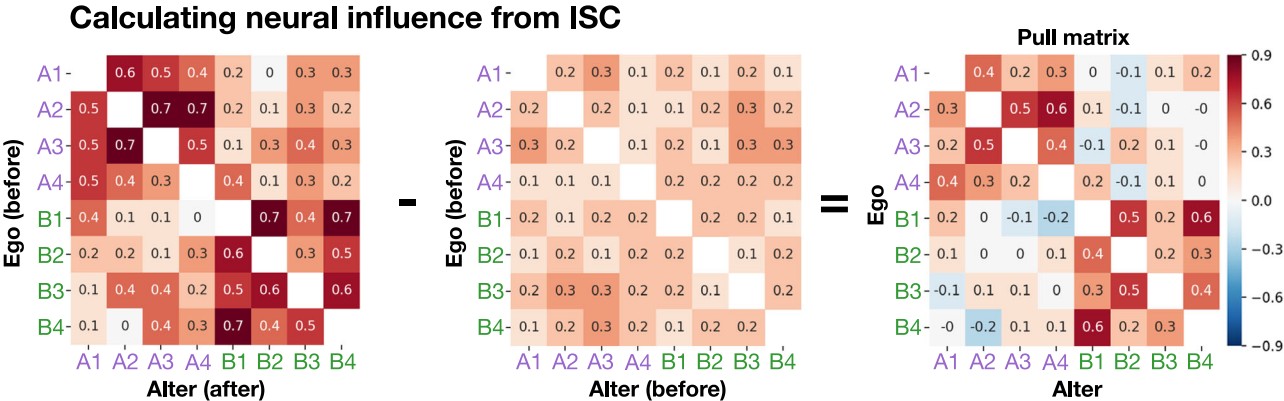

**Fig. 6 | Neural influence analysis example.** A worked example of how neural influence was calculated based on ISC, for two groups each containing four participants. The *influence matrix* is determined by assessing how far the alter moved toward the ego's initial position.

the target, participant ID as the predictor, and random intercepts for rater IDs.

At the conversation group level ($n = 45$ group–movie pairs, 9 conversation groups each watching 5 movies), exploratory multi-model inference was performed using the R package `MuMIn`[86], selecting the regression models that best predicted changes in brain alignment and changes in answers to survey questions. The following predictors and their interactions were included: the total number of words spoken by the group[59,87,88], as well as the inequality across group members of words spoken, PCA centrality, and perceived social status, each measured using the Gini coefficient[89]. These predictors were selected to capture leader–follower dynamics[90–93] and because previous research indicates that high-centrality participants may have specific conversational skills[25–27]. Separate hierarchical models were fit for each alignment DV, with random intercepts for movie clips. Models were fitted using the `lme4` package for R[94]. Because the `MuMIn` package

is limited to a maximum of 30 predictors (including interactions), it was not possible to include both the central tendencies and Gini coefficients of PCA centrality and perceived status. The best model fits with significant individual predictors were obtained using only Gini coefficients, so the medians of PCA centrality and perceived status were excluded. The central tendency of PCA centrality was negatively correlated with its Gini coefficient ($R = -0.61$, $p < 0.001$), so this analysis cannot distinguish higher median centrality from lower inequality of centrality. See Supplementary Table 1 for regression statistics.

At the speech turn level ($n = 16,057$ speech turns) our focus was on whether individual variables predicted perceived status and PCA centrality, and not on total variance explained. Accordingly, a separate model was trained for each continuous variable, ensuring variance shared between predictors was represented in the results. For these continous variable models, *P*-values were corrected for multiple comparisons using Holm's[95] sequential procedure. Types of speech

turns (see above) were treated differently: Because speech turn types were mutually exclusive, a single model was trained for each dependent variable with type as a categorical predictor, and predictor $P$-values were not corrected for multiple comparisons. Both continuous and categorical models were controlled for the number of words spoken by participants and included random intercepts for speaker and coder identities to account for the hierarchical structure of the data. Analysis at the conversation level included a small number of predictors, so a single hierarchical model with all predictors and interactions was trained, with random intercepts for group identity. Models were fitted using the `lme4` package for R[94]. See Supplementary Tables 3 and 5 for regression statistics.

At the word level ($n = 8122$ speech turns), all conversation transcripts were preprocessed using Gensim[96]. This included stripping punctuation, white-space, and numeric values, removing short words and a standard list of stop-words, and word stemming using Porter's[97] algorithm. Word stems used less than 5 times across all conversations were replaced with the placeholder "RAREWORD." To partially account for the hierarchical structure of the data, word stems were only included in further analysis if they were used by 10 or more participants and in discussions of 3 or more movie clips, excluding participant– and movie– specific terms. 352 word stems were used in the final analysis. A matrix was constructed where rows corresponded to speech turns and columns corresponded to word stem usage counts. Columns were added for the speaker's z-scored PCA centrality, and z-scored perceived status. Using the `statsmodels` package for Python[98], separate ordinary least squares regressions were performed with PCA centrality and perceived status as target variables and word stem counts as predictors. Beta values are interpretable as predicted increases in the number of standard deviations from the network-wide mean PCA centrality and experiment-wide perceived status values. See Supplementary Table 2 and Supplementary Data 2 for word-level regression statistics.

### Marginal $R^2$
For mixed-effects models, marginal $R^2$ captures the variance accounted for by fixed-effects only and was calculated using a modified version of the procedure of Nakagawa & Schielzeth[99], as implemented by the R library `MuMIn`[86] or directly using the Python package NumPy. The marginal $R^2$ $P$-value was calculated using permutation testing: The analysis was repeated with at least 2000 different random permutations of the DV to create a null distribution of marginal $R^2$ values, and the $P$-value was the fraction of the null values as or more extreme than the empirical marginal $R^2$ value.

### Assumptions of statistical tests
All $P$-values specify whether they were calculated parametrically or with permutation testing. All brain maps used permutation tests and were corrected for multiple comparisons. When permutation tests were used, the data and randomization procedures satisfied the assumption of exchangeability. Parametric $P$-values were used when permutation tests were impractical (e.g., when mixed-effects models with large sample sizes failed to converge on randomly permuted data). For the parametric $P$-values, tests of normality of residuals and equal variance were not reported because even very small deviations from normality are significant at large sample sizes, and because mixed-effects models used are considered robust to violations of these assumptions[100].

### Multidimensional scaling
Multidimensional scaling (MDS) plots in Supplementary Material show a 2-dimensional projection of the relative distances between participants BOLD time series, before and after conversation. MDS was performed using Scikit-learn[85] using the correlation distance between participants' BOLD time series.

### Image credits
"Brain" icon by Clockwise from Noun Project, availble at https://thenounproject.com/icon/brain-1080481/. "Meeting" icon by SBTS from Noun Project, available at https://thenounproject.com/icon/meeting-5279011/. "Clipboard" by Made by Made from Noun Project, available at https://thenounproject.com/icon/clipboard-674066/. "Film" icon by NeueDeutsche from Noun Project, available at https://thenounproject.com/icon/film-531914/.

### Reporting summary
Further information on research design is available in the Nature Portfolio Reporting Summary linked to this article.

### Data availability
The raw fMRI data generated in this study have been deposited in the NIMH Data Archive database under collection ID #771, available at https://doi.org/10.15154/1504150[101]. Stimuli, anonymized conversation transcripts, conversation ratings, social network analysis derivatives, and fMRI analysis derivatives are available in the Open Science Framework database, available at https://osf.io/kr9fb/[102]. The raw social network data are protected and are not available due to data privacy laws.

### Code availability
Preprocessing code, analysis code, and figure rendering code are available in the Open Science Framework database, available at https://osf.io/kr9fb/[102].

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

## Acknowledgements

We thank Robert Burnham developing the scheduling tool, Paulina Calcaterra for assistance collecting data, Margaret Lawson for developing software used during piloting, Caitlyn Lee, Bryn Williams, Ben Swett, Krista Schemitsch, Elizabeth Garvey, Elizabeth Adams, Kai Lord, Catherine Gorman, Halla Hafermann, Hwikeun Kim, Jeffrey Cho, Justin Chang, Kelly Shin, Sarah Yoo, Tobias Choyt, and Toby Lee for assistance evaluating the conversations, and Jamil Zaki, Karen Huang, Luke Chang, Jeremy Manning, Janice Chen, Chris Honey, Josh Greene, Peter Tse, and Jim Haxby for insightful discussion. Project funded by the NIMH of the National Institutes of Health under award number R01MH112566-01 to T. Wheatley and U. Hasson, and Dartmouth seed funding to T. Wheatley and A. Kleinbaum.

## Author contributions

B.S. Conceptualization, data curation, formal analysis, investigation, project administration, software, visualization, writing - original draft, review & editing. C.W. Investigation, writing - review & editing. U.H. Conceptualization, funding acquisition, supervision, writing - review & editing. A.K. Conceptualization, funding acquisition, investigation, resources, supervision, writing - review & editing. T.W. Conceptualization, funding acquisition, resources, supervision, writing - review & editing. Roles defined by the Contributor Roles Taxonomy, available at https://casrai.org/credit/.

## Competing interests

The authors declare no competing interests.
