## [Peer Review File · Nature Communications]

Consensus-building conversation leads to neural alignmentREVIEWER COMMENTS

Reviewer #1 (Remarks to the Author):

This is potentially a most exciting study reporting fascinating results. It shows how discussion can lead to consensus reflected in alignment of brain activity. However, when I tried to examine the details I became increasingly concerned. The main problem is that much important information is not in the main text, but in the supplementary material. I would like to see more information in the main text: numbers of subject, size of discussion groups, &c. I also found that much of the statistical analysis is outwith my expertise. More expert advice will be needed.

1) 49 subjects were scanned, including a control group of 9. 9 is a very small number for a brain imaging study.

2) The control group is very important for demonstrating that the group discussions caused the alignment. I was unable to find any direct statistical comparison of the control group with the experimental group, although there was a brief mention that the control group also showed some increased alignment.

3) Participants' survey answers became more similar after discussion. I would like to know more about this. For example, how similarity was measured. Also, I don't understand where the df of 1269 comes from (bottom of page 3), given that there were 9 groups of ~4 people.

4) The data about the social networks is most interesting, but, as the authors admit, somewhat exploratory. The correlations reported are extremely low; e.g. for fig 1, $r=0.09$. The graph suggests that there was a relationship, but where does $N=3478$ come from given that there were 865 participants. The lower graph in fig 3 is not in the least convincing.

This is an exciting project, so I hope these problems can be resolved.

Reviewer #2 (Remarks to the Author):

This paper examines changes in fMRI inter-subject correlation (ISC) observed during film clips, before and after groups conversations about the meaning of the clips. The paper presents evidence that ISC increased after the conversation (for both the previously viewed clips and related clips from the same film). It also relates individual differences in the degree of conversation-related ISC change to centrality in a social network.

Overall, this is an interesting paper describing results from a fascinating and creative project. However, I wasn't able to understand all of the results to the extent that I would have liked to. Below I include comments and questions for the authors to consider, which are intended to help them improve the presentation of the work.

Comments:

- It was difficult for me to understand the nature of the analyses as they are presented here, and how they connect with inferential goals of this investigation. For example, I would like to understand the statement on the bottom of page 3 that "A bilateral set of brain regions exhibited conversation-related change in ISC, supporting the hypothesis that consensus-driven conversation can align future brain activity", together with how it relates to Figure 1 and the explanation on page 5 through 7 of the Supplementary Materials PDF. Here are some questions related to this that I think could be more

clearly explained: What is the regression that was fit at each voxel (e.g., code, examples of full regression table output) ? What was the number of datapoints / degrees of freedom for these regressions? How was the multilevel nature of the dataset accounted for (if at all)? How was the property "proportion of significant movie-group combinations" from Figure 1 constructed and what is this meant to communicate? How is this property "proportion of significant movie-group combinations" relevant to the inference that a nonzero change in ISC was causally evoked by the between-group conversation manipulation? Possibly this is all straightforward but I wasn't quite able to gather it from the manuscript.

- Re: Figure 2 -- what is the regression model that was fit here? What is the outcome variable that was explained via these components (is it change in correlation?)? What are the datapoints (voxels? regions?) ? What are the values of the predictor variables (posterior probabilities from Neurosynth for each voxel/region?)

- Re: Figure 3 -- same comment as above in that what actual models were fit is not clear to me (perhaps it would be useful to report the models that were fit, with summary regression tables, as a supplement, to the extent that this is possible). I am also confused about the number of datapoints going into these models and what the means about the inferences drawn -- e.g., are egoID, alterID, and egoID:alterID all treated as fixed factors? The scatterplot in Figure 2 -- what is each individual aggregated datapoint (with interval) meant to reflect/communicate?

- Re: the inference that high-centrality participants facilitated consensus -- I am also confused about this part. For Supplementary Figure 11, what are the variables that are presented on the x axes? What are the variables presented on the y axes? What is each datapoint? If each datapoint is an individual person, is the finding presented that individual differences in the degree of change (in behavioral responses or in ISC across a variety of brain regions) are correlated with individual differences in the product of their group's Gini coefficient and their group's number of words spoken? Given this multilevel structure, what is the effective degrees of freedom for estimating this relationship? Why are so many datapoint assigned a value of exactly zero for "Neural change"? What is meant by the property "Groups with no significant localized neural changes" referred to in the figure caption, and how can this property/idea justify the assignment of the value of zero to these datapoints?

Reviewer #3 (Remarks to the Author):

NCOMMS-20-35897-T "How consensus-building conversation changes our minds and aligns our brains."

This paper demonstrates neural convergence among participants after discussing ambiguous content. Participants viewed ambiguous movie clips, discussed what they saw, and then viewed subsequent clips from the same movies. That participants neural activity became similar after discussion is taken as evidence of synchronization among participants (relative to the control). Further, evidence suggests that more central participants (in their MBA friendship network) adapt more and function as a sort of glue for the group. All in all, this work contributes, in multiple ways, to interdisciplinary work on influence and network science, which makes Nature Comm a reasonable venue.

The main shortcoming of this work, in my opinion, is that the discussion phase is a black box. Word count is modeled, but the dynamics of the discussion, which seem to be very important for consensus building, are ignored. We know that participation rates, social influence, and evaluations of participation are not evenly distributed. Higher status people (either exogenously defined by things like position, race or gender; or endogenously defined in terms of having an influence attempt be

accepted or receiving a positive evaluation) guide the interaction, are influential and receive praise for their efforts. Why are the interactions not coded to determine a pecking/status order for the group? This information is poorly proxied by the centrality scores, but could be directly recovered from the transcripts. That, to me at least, would be more powerful – if you could point to interaction patterns that result in greater ISC.

Considering the importance of consensus building, and the fact that there were only 9 groups total, I'd like to see a graph showing the consensus building for each of the 9 groups. Descriptively, how much change was there in their opinions and in the ISC score? This seems like an important bit of basic information with which to lead.

More minor points to consider:

Much of the front end in this paper reminded me of Deutsch and Gerard's (1955 "A Study of Normative and Informational Influences..." *Journal of Abnormal and Social Psychology* 51(3): 629-36) distinction between normative and informational influence. Informational influence occurs when someone changes their mind in light of evidence, while normative influence occurs when someone says something to go along with a group, but whether they believe it is unknown. This distinction is relevant when the author state, "[those] offering explanations of social influence have therefore often treated conversation as a means of achieving public compliance, setting internal beliefs aside." Or "However, it remains unclear whether these conversations shape the way individuals fundamentally see the world, or are merely a means of enforcing public compliance." This latter quote is directly conceptualizing influence in terms of normative or informational influence. The methods outlined in this paper seem to be a good way to establish whether something is normative or informational, but given the importance of this distinction, I think the authors should address this notion.

With respect to the network centrality results, there is an obvious chicken-and-egg problem that the authors do not address. Are central members central because they are more neurally flexible, or are they neurally flexible because they're central? I think point is important enough to warrant some discussion.

You make the point that brain alignment may shape who befriends whom (page 2). On this topic, you might consider similar work on neural correlates of reciprocity. For example: Zerubavel et al. 2018. *PNAS* 115(17):4375-4380.

The sample size is quite small (by my sociological standards), but I recognize that this varies from field to field (but by psychological standards, why is there only one study?). The actual sample size is only available in the SI. It should be included in the results so that when you say "All but two..." agreed with the consensus, we have a sense of what that means relative to the sample space.

It's unclear to me why Brokerage was used at all. Centrality also isn't justified as a measure, but it is a reasonable "default" network measure to explore associations between network measures and neural change. It seems some motivation for these measures is warranted. Why would centrality, or brokerage, or any other network statistic make sense here? More to the point, are there any theories that suggest some relational mechanism that is tapped by a network measure? The reason I bring this up is that the social network component of this comes off as a mere add-on, that you included only because you had the measures. There is very little, if any, "theory" guiding all of this work, and that makes the connection to social networks very tenuous.

P. 10 of the SI: you state that you simultaneously modeled "centrality, eigenvector centrality, and brokerage..." Is this a typo or is there another operationalization of centrality aside from eigenvector? Also, does including all three result in collinearity? They should be pretty highly correlated measures.

Response to reviewers

We thank the editor and reviewers for the helpful feedback. Wherever possible, we have revised the manuscript to follow the reviewers' recommendations. As requested by Reviewer 3, **we have conducted new analyses of the conversation content**. These analyses are reported in the section *Results: Network centrality and perceived social status affect group alignment*. The discussion section has also been revised in light of these new results. Performing the conversation content analyses required managing a team of raters that read and classified every speech turn of every conversation. Because of the ongoing pandemic, this process took longer than we anticipated, and we are thankful for the editor and reviewers' patience. Additionally, we have made minor changes to improve clarity.

Below, we respond to each reviewer in detail, describing the revisions made to address their comments. Our responses are indented and in red, with relevant excerpts from the revised manuscript block-quoted and in a smaller typeface. All references are available in the revised manuscript.

Reviewer 1:

This is potentially a most exciting study reporting fascinating results. It shows how discussion can lead to consensus reflected in alignment of brain activity.

We thank the reviewer for their kind comments.

However, when I tried to examine the details I became increasingly concerned. The main problem is that much important information is not in the main text, but in the supplementary material. I would like to see more information in the main text: numbers of subject, size of discussion groups, &c. I also found that much of the statistical analysis is outwith my expertise. More expert advice will be needed.

This manuscript was directly transferred to Nature Communications from another Nature journal that required the Methods be in an online-only supplementary section, outside of the main text. We have revised the manuscript to move the Methods section back into the main text.

1) 49 subjects were scanned, including a control group of 9. 9 is a very small number for a brain imaging study. The control group is very important for demonstrating that the group discussions caused the alignment. I was unable to find any direct statistical comparison of the control group with the experimental group, although there was a brief mention that the control group also showed some increased alignment.

The reported results were controlled by the inclusion of an intercept term in the regression model that captures changes in neural alignment in all participant pairs regardless of group membership, not by direct comparison with the control group. We took this approach

because it allowed us to fit a statistical model including all groups, following the recommendation of G. Chen et al. (2016). Appropriately, the main findings do not depend on between-group post hoc tests. The manuscript now addresses this issue as follows:

Results, footnote 1: fMRI regression analyses were controlled by an intercept term corresponding to all within- and between-group participant pairs, including control group participants. This intercept term captured the effect of watching the movie clips twice, with or without conversation, regardless of conversation group. Unless otherwise noted, the reported results are relative to this intercept. Some localized positive changes in ISC were observed for the control group. This may be because some information was shared within most conversation groups, but not among controls, or because simply having a conversation, regardless of content, changed how participants engaged with the movie clips. See *Methods: Conversation-induced change in ISC*.

3) Participants' survey answers became more similar after discussion. I would like to know more about this. For example, how similarity was measured.

We have revised the manuscript to state how similarity was measured:

Results, line 128: Similarity was measured using city block distance.

Also, I don't understand where the df of 1269 comes from (bottom of page 3), given that there were 9 groups of ~4 people.

The df value for this t-test is large because it compares distances between pairs of participants. There were 61 within-group pairs and 1210 between-group pairs, giving an independent samples t-test df value of $61 + 1210 - 2 = 1269$. We have revised the manuscript to clarify this issue, using more specific labeling and adding Ns for each sample:

Results, line 124: [...] after conversation, participants' survey answers became more similar to the answers of their conversation group members, compared to those in other groups (within-group participant pairs: $N=61$, mean distance=4.61, 95% CI=[3.9, 5.31]; between-group participant pairs: $N=1210$, mean distance=8.28, 95% CI=[8.11, 8.44]; $t(1269)=-10.08$, $p<.001$).

4) The data about the social networks is most interesting, but, as the authors admit, somewhat exploratory. The correlations reported are extremely low; e.g. for fig 1, $r=0.09$. The graph suggests that there was a relationship, but where does $N=3478$ come from given that there were 65 participants.

As with the large df value described above, the large N for this correlation is accounted for by the number of unordered pairs of participants both within and between groups. 47 participants make 1081 pairs, times 5 runs gives 5405. Because we could not calculate behavioral change for all participants for all runs due to missing data, some pairs had to be excluded (described in *Exclusions and missing data*, line 392), giving an N of 3478 pairs for this correlation. A related analysis on neural and behavioral influence uses ordered pairs, giving an N of 8620. We have revised the manuscript to state these units of analysis explicitly:

Results, line 147: Participant pairs whose survey answers became similar showed greater whole-brain alignment, across movie clips and groups ($R=0.09$, $p<.001$, $N\text{ pairs}=3478$), confirming that neural alignment was driven in part by change in belief (see *Supplementary Material: Comparing behavioral and neural results*).

Results, line 215: Confirming that neural influence was related to changes in belief, survey influence calculated over the survey answers of ordered participant pairs was correlated with whole-brain neural influence ($R=0.06$, $p<.001$, $N\text{ pairs}=8620$) (see *Supplementary Material: Comparing behavioral and neural results*).

Supplementary Material, Comparing behavioral and neural results: [...] This analysis was conducted over all unordered pairs of participants where meaning vector change could be calculated across all five runs. [...] This analysis was conducted over all ordered pairs of participants where influence could be calculated across all five runs.

The lower graph in fig 3 is not in the least convincing.

Figure 3 visualizes a small but statistically significant effect. We have retained the lower graph in Figure 3 because we want to give an accurate picture of how this small effect emerges from the data. We have included additional information to aid interpretation of the figure:

Figure 3, caption: Dots represent the mean mass of the neural influence map across participant pairs where the ego (top) or alter (bottom) had the indicated PCA centrality value, where the pair conversed in the same group and where complete behavioral data were available. Vertical lines show 95% confidence intervals. Diagonal lines and shaded regions show regression lines of best fit and their 95% confidence intervals.

This is an exciting project, so I hope these problems can be resolved.

Reviewer #2:

This paper examines changes in fMRI inter-subject correlation (ISC) observed during film clips, before and after groups conversations about the meaning of the clips. The paper presents evidence that ISC increased after the conversation (for both the previously viewed clips and related clips from the same film). It also relates individual differences in the degree of conversation-related ISC change to centrality in a social network.

Overall, this is an interesting paper describing results from a fascinating and creative project. However, I wasn't able to understand all of the results to the extent that I would have liked to. Below I include comments and questions for the authors to consider, which are intended to help them improve the presentation of the work.

We appreciate the reviewer's kind comments, and hope the revisions described below address their concerns.

Comments:

- It was difficult for me to understand the nature of the analyses as they are presented here, and how they connect with inferential goals of this investigation. For example, I would like to understand the statement on the bottom of page 3 that "A bilateral set of brain regions exhibited conversation-related change in ISC, supporting the hypothesis that consensus-driven conversation can align future brain activity", together with how it relates to Figure 1 and the explanation on page 5 through 7 of the Supplementary Materials PDF.

Here are some questions related to this that I think could be more clearly explained: What is the regression that was fit at each voxel (e.g., code, examples of full regression table output)?

We have revised the manuscript to explain the regression model more clearly and to indicate that the Supplementary Material contains a full table of fMRI results. See also Figures 5 and 8, which demonstrate how the fMRI analysis predictor matrices were constructed.

Results, line 139: Conversation-related change in ISC was calculated by subtracting ISC before conversation from ISC after conversation, for each pair of participants. Multiple regression with multiple comparisons correction was used to localize change in ISC unique to each movie–group combination, over and above the effect of simply watching the movie clips twice. [See footnote 1, reproduced below.] A permutation testing method that accounted for the grouped structure of the data was used to limit the false positive rate (G. Chen et al., 2016). Complete regression results are listed in Supplementary Table 5. The scope of inferential statistical generalization is limited to the specific movie clips and groups reported here. See *Methods: Conversation-induced change in ISC* and *Supplementary material: Limitations on generalization*.

Results, footnote 1: fMRI regression analyses were controlled by an intercept term corresponding to all within- and between-group participant pairs, including control group participants. This intercept term captured the effect of watching the movie clips twice, with or without conversation, regardless of conversation group. Unless otherwise noted, the reported results are relative to this intercept. Some localized positive changes in ISC were observed for the control group. This may be because some information was shared within most conversation groups, but not among controls, or because simply having a conversation, regardless of content, changed how participants engaged with the movie clips. See *Methods: Conversation-induced change in ISC*.

Methods: Conversation-induced change in ISC, line 506: For each fMRI session, ISC was calculated for every participant pair, creating an ISC matrix. A *change* matrix was calculated by subtracting before-conversation ISC from after-conversation ISC. The effect of group membership on change in ISC was quantified using regression analysis. Predictor matrices were created for each group, with 1s where both participants were in the target group and 0s otherwise. An all-1s intercept matrix was used to capture change that occurred whether or not participants were in the same group (i.e., change caused simply by watching a movie clip twice). Unraveled vector versions of these matrices were used as predictors in a multiple regression with the unraveled change matrix as the target, executed using NumPy `linalg.lstsq()`. The beta values from this regression represent the change in ISC unique to each group. See Figure 5 for a worked example. Repeating this procedure for each movie clip across every voxel in the fMRI data yielded *change maps* showing where in the brain change occurred for each movie–group combination. See *Results: Conversation aligned future brain activity* and Figure 1.

What was the number of datapoints / degrees of freedom for these regressions?

We have revised the regression results table caption to include the degrees of freedom for each analysis, as well as to more clearly indicate that it contains regression results for each region of interest.

Supplementary Material, Supplementary Table 5: Significant clusters for all whole-brain regression analyses after correction for multiple comparisons ($p < .05$ corrected). The mean, peak, and 95% CI columns describe beta estimates for the indicated movie–group combination and analysis. See *Methods* for complete descriptions of each analysis. For the neural influence analysis, the group column additionally specifies whether ego or alter centrality was mapped and which centrality measure was used (EC: eigenvector centrality; br.: brokerage; cent.: PCA centrality). Degrees of freedom for each analysis: Change df = 1070; Influence df = 2150; Unseen df = 1070.

How was the multilevel nature of the dataset accounted for (if at all)?

The multilevel nature of the dataset was partially accounted for using a permutation-testing method described by G. Chen et al. (2016). This approach appropriately limits the false positive rate given the pair-and-group structure of the analysis, though is not equivalent to the use of a full multilevel model. Revising the fMRI analysis pipeline to use a full multilevel model would require substantial additional time investment and would not be likely to change the conclusions of the paper, which do not depend on estimating random effects for movies and groups. We have, however, revised the manuscript to highlight the limitations of the permutation testing approach not only in the Supplementary Materials, but also in the main text Results and Methods sections:

Results, line 142: A permutation testing method that accounted for the grouped structure of the data was used to limit the false positive rate (G. Chen et al., 2016). The scope of inferential statistical generalization is limited to the specific movie clips and groups reported here. See *Methods: Conversation-induced change in ISC* and *Supplementary material: Limitations on generalization*.

Methods, Conversation-induced change in ISC, line 521: Significance values were calculated using subject-wise permutation testing (G. Chen et al., 2016). The regression described above was repeated 2000 times. Each time, the rows and columns of the change matrix were identically permuted (Figure 5, bottom). This procedure satisfies the exchangeability assumption of permutation testing, appropriately limiting the false positive rate. Two-tailed p-values for each parameter were obtained by calculating the proportion of permutations where the true result was more extreme than the permuted results. Statistical significance was achieved when the true result was more extreme than 97.5% of the permuted results, establishing a two-tailed alpha value of 0.05. This approach controls the false positive rate given the pairs-and-groups structure of the study, but it is not equivalent to the use of a multilevel model. See *Supplementary Material: Limitations on generalization*. Multiple comparisons correction was performed at the cluster level using AFNI 3dClustSim. Cluster simulation used a non-Gaussian ACF model allowing for heavy tails, using model parameters estimated from the mean of the collected fMRI data using AFNI 3dFWHMx

Supplementary Material, Limitations on generalization: [...] movie clips and groups were treated as fixed effects, limiting the scope of inferential statistical generalization to the specific movie clips and groups reported here; different groups discussing different movies may exhibit different patterns of alignment

and influence. Comprehensive assessment of the conditions under which conversation does or does not align neural activity remains an important avenue for future research.

How was the property "proportion of significant movie-group combinations" from Figure 1 constructed and what is this meant to communicate? How is this property "proportion of significant movie-group combinations" relevant to the inference that a nonzero change in ISC was causally evoked by the between-group conversation manipulation? Possibly this is all straightforward but I wasn't quite able to gather it from the manuscript.

We predicted that change in alignment would occur in different parts of the brain for different conversations about different movies. For each movie–group combination, regression analysis produced a change map showing where in the brain change occurred. This regression directly tests the hypothesis that conversation causes change in ISC and the results are fully reported in the Supplementary Material. The proportion of significant movie–group combinations is the number of significant movie–group combinations divided by the total number of movie–group combinations at each voxel. This is a simple and interpretable way to summarize the regression results, showing where in the brain change tends to occur across movie clips and groups. We have revised the manuscript to clarify:

Results, line 129: A bilateral set of brain regions exhibited conversation-related change in ISC, supporting the hypothesis that consensus-driven conversation can align future brain activity (see *Supplementary Material: Cluster reports* and Supplementary Table 5 for all fMRI analysis results). Alignment changes took place in different brain areas for different groups discussing different movie clips. Because alignment depends on features of both the group and the conversation, and because conversing groups were small, some brain areas survived multiple comparisons correction across a greater proportion of movie–group combinations (Figure 1; Supplementary Figure 2). Alignment tended to increase in visual and auditory sensory areas, as well as in higher-order areas associated with the attention and default mode networks, including the temporal parietal junction, angular gyrus, posterior cingulate, medial prefrontal cortex, and temporal pole.

- Re: Figure 2 -- what is the regression model that was fit here? What is the outcome variable that was explained via these components (is it change in correlation?)? What are the datapoints (voxels? regions?) ? What are the values of the predictor variables (posterior probabilities from Neurosynth for each voxel/region?)

We have revised the manuscript to clarify that we used Principal Component Analysis to identify terms that tended to occur together across all reported fMRI results, to state that additional detail on the Neurosynth analysis is included in the Methods section, and to state that the posterior probabilities for each brain region are included in the Supplementary Materials:

Results, line 160: Conversation aligned brain areas associated with a wide range of cognitive processes, as estimated by quantitative reverse inference using Neurosynth (Yarkoni, Poldrack, Nichols, Van Essen, & Wager, 2011). Neurosynth uses a large database of previously published brain imaging literature, where English-language terms are associated with brain activation likelihood maps. For each statistically significant brain area in all reported fMRI analyses, we estimated which Neurosynth terms were a likely match. We then identified groups of terms that tended to co-occur using principal component analysis. 18

groups of terms explained 70% of the variance in the term probability data (Figure 2). The group of terms that explained the most variance included words related to vision and motion, suggesting that one important function of conversation is future alignment of visual attention. Other high-ranking principal components were associated with motor activity, working memory, face perception, the default network, auditory perception, body movement and social understanding, and language. See *Supplementary Material: Cluster reports* for term probabilities for each brain region and *Methods: Reverse inference with Neurosynth* for additional detail.

Methods: Reverse inference with Neurosynth, line 588: For each significant region in all of the foregoing analyses and each of the 3,228 Neurosynth terms, we obtained a vector of probabilities that a term would be used, given the spatial pattern of the result (i.e., the posterior probability(feature|activation), assuming an empirical prior). This yielded a matrix where rows corresponded to brain regions and columns to term posterior probabilities. Anatomical and duplicate terms were removed. The columns of the term probability matrix were reduced to 18 components using Principal Component Analysis, capturing 70% of the variance in the data.

- Re: Figure 3 -- same comment as above in that what actual models were fit is not clear to me (perhaps it would be useful to report the models that were fit, with summary regression tables, as a supplement, to the extent that this is possible).

We have revised the manuscript to explain the regression model more clearly, and to indicate that the Supplementary Material contains a full table of fMRI results. See also Figure 8, which contains a worked toy example demonstrating how the influence matrices were created.

Results, line 208: Neural influence occurred in different parts of the brain for different pairs of participants. To account for these differences, multiple regression was used to localize network centrality-dependent neural influence in specific brain areas. See *Methods: Neural influence* for a detailed description of the regression model and Supplementary Table 5 for all fMRI analysis results. Centrality-dependent influence affected a broad set of brain regions for both egos and alters, showing that high-centrality participants were both more neurally influential and more likely to be neurally influenced by others in some brain areas (Figure 3; Supplementary Figures 3 and 4).

Methods: Neural influence, line 566: We refer to the participant hypothesized to exert neural influence as the *ego*, and the participant hypothesized to be influenced as the *alter*. First, ISC matrices including every possible ego–alter pair were calculated. The *initial position matrix* measured the ISC of the ego and the alter during the before-conversation fMRI session, while the *final position matrix* measured the ISC of the ego before conversation with the alter after conversation. To obtain an *influence matrix*, the initial position matrix was subtracted from the final position matrix (Figure 8). Unlike the change analysis above, in which only the lower triangle of each matrix was used, in the influence analysis both the upper and lower triangles were used, to test both members of each pair in the ego *and* the alter position. This procedure yielded neural influence maps at the participant-pair level, with values directly interpretable as how much more similar the alter became to the ego's initial time series.

Methods: Neural influence, line 576: The effect of social network centrality was assessed using regression analysis over the neural influence maps generated at the level of the participant pair. Predictor matrices were structured similarly to the predictors used in the change analysis, but instead of 1s, each cell included the social network metric values of either the ego or the alter. Because predictor beta estimates made with ordinary least squares regression depend only on the variance unique to that predictor, this approach allowed us to separately estimate the effects of PCA centrality, eigenvector centrality, and

brokerage on neural influence. This regression yielded maps of centrality-dependent neural influence unique to each movie–group combination.

I am also confused about the number of datapoints going into these models and what the means about the inferences drawn -- e.g., are egoID, alterID, and egoID:alterID all treated as fixed factors?

We appreciate the reviewer's concerns about how to account for the multi-level structure of the data. In our response above we clarified that the multi-level structure of the data was partially accounted for using subject-wise permutation testing (G. Chen et al., 2016). This approach appropriately controlled the false positive rate but did not allow us to estimate separate random slopes and intercepts for participant IDs and their interactions. The manuscript does not claim to estimate such effects, and directly addresses how the use of fixed effects places limits on statistical generalization:

Supplementary material, Limitations on generalization: Finally, movie clips and groups were treated as fixed effects, limiting the scope of inferential statistical generalization to the specific movie clips and groups reported here; different groups discussing different movies may exhibit different patterns of alignment and influence.

The scatterplot in Figure 2 -- what is each individual aggregated datapoint (with interval) meant to reflect/communicate?

We have revised the manuscript to clarify the meaning of the dots in the scatterplots of Figures 1 and 3:

Figure 1, caption: Dots represent the mean mass of the neural change map across participant pairs with the indicated survey change value, where complete behavioral data were available. Vertical lines show 95% confidence intervals. Diagonal line and shaded region show the regression line of best fit and its 95% confidence interval.

Figure 3, caption: Dots represent the mean mass of the neural influence map across participant pairs where the ego (top) or alter (bottom) had the indicated PCA centrality value, where the pair conversed in the same group and where complete behavioral data were available. Vertical lines show 95% confidence intervals. Diagonal lines and shaded regions show regression lines of best fit and their 95% confidence intervals.

- Re: the inference that high-centrality participants facilitated consensus -- I am also confused about this part. For Supplementary Figure 11, what are the variables that are presented on the x axes? What are the variables presented on the y axes? What is each datapoint? If each datapoint is an individual person, is the finding presented that individual differences in the degree of change (in behavioral responses or in ISC across a variety of brain regions) are correlated with individual differences in the product of their group's Gini coefficient and their group's number of words spoken?

We appreciate the reviewer's concern about the visualization of two- and three-way interactions. Reviewer 3 recommended that we undertake a more extensive analysis of conversation content, as well as perform an analysis that better accounted for the hierarchical nature of the data. In doing so, we have replaced this analysis completely. See the sections *Results: Social status and centrality affect group alignment* and *Methods: Conversation content analysis*, which contain several new analyses. These analyses are also described in brief at the top of this document.

Given this multilevel structure, what is the effective degrees of freedom for estimating this relationship?

We have revised the manuscript to describe how the multilevel structure of the data was accounted for by the new analyses.

Methods: Conversation content analysis, line 624: To account for the hierarchical structure of the data, analyses at the speech turn level included random intercepts for speaker, group, movie, and coder identities. Because our focus was on individual parameters, as opposed to total variance explained, a separate model was trained for each predictor, ensuring variance shared between predictors was represented in the results. P-values were corrected for multiple comparisons using Holm's (1979) sequential procedure. Analysis at the conversation level included a small number of predictors, so a single hierarchical model with all predictors and interactions was trained, with random intercepts for group identity. Models were fitted using the lme4 package for R (Bates, Mächler, Bolker, & Walker, 2015). See Supplementary Table 2 for regression statistics including estimated degrees of freedom.

Methods: Conversation content analysis, line 637: To partially account for the hierarchical structure of the data, word stems were only included in further analysis if they were used by 10 or more participants and in discussions of 3 or more movie clips, excluding participant- and movie- specific terms.

Why are so many datapoint assigned a value of exactly zero for "Neural change"? What is meant by the property "Groups with no significant localized neural changes" referred to in the figure caption, and how can this property/idea justify the assignment of the value of zero to these datapoints?

The datapoints assigned a value of exactly zero were movie-group combinations where localized changes in neural alignment did not survive correction for multiple comparisons. We retained these datapoints in the regression and the visualization to emphasize that the analysis did not exclude any data.

Reviewer #3:

This paper demonstrates neural convergence among participants after discussing ambiguous content. Participants viewed ambiguous movie clips, discussed what they saw, and then viewed subsequent clips from the same movies. That participants neural activity became similar after discussion is taken as evidence of synchronization among participants (relative to the control). Further, evidence suggests that more central participants (in their MBA friendship network) adapt more and function as a sort of glue for the group. All in all, this work contributes, in

multiple ways, to interdisciplinary work on influence and network science, which makes Nature Comm a reasonable venue.

The main shortcoming of this work, in my opinion, is that the discussion phase is a black box. Word count is modeled, but the dynamics of the discussion, which seem to be very important for consensus building, are ignored. We know that participation rates, social influence, and evaluations of participation are not evenly distributed. Higher status people (either exogenously defined by things like position, race or gender; or endogenously defined in terms of having an influence attempt be accepted or receiving a positive evaluation) guide the interaction, are influential and receive praise for their efforts. Why are the interactions not coded to determine a pecking/status order for the group? This information is poorly proxied by the centrality scores, but could be directly recovered from the transcripts. That, to me at least, would be more powerful – if you could point to interaction patterns that result in greater ISC.

We thank the reviewer for their comments, which pushed us to complete new analyses of the group conversations. In part due to the pressure of the pandemic, these analyses took a long time, and we appreciate the patience of the reviewers and the editor. The present revision of the manuscript has been extensively revised to describe the new analyses and results in detail:

Abstract, line 19: Groups with participants perceived as having high social status showed less alignment, while groups with participants who were central in their real-world social networks showed greater alignment. High-status participants signaled disbelief in others' proposals and spoke more, producing unequal turn-taking within their groups. By contrast, high-centrality participants encouraged others to speak, producing more equal turn-taking.

Introduction, line 44: Further, we found that some conversations created more alignment than others. Groups with participants perceived as having high social status showed unequal turn-taking and much lower alignment, whereas groups with participants who were more central in their social networks showed more equal turn-taking and greater alignment. This may be because perceived high-status participants signaled disbelief in others' proposals and spoke more, disrupting group consensus. By contrast, high-centrality participants facilitated equal turn-taking and were more likely to adapt their own brain activity to the group.

1. *Results: Network centrality and perceived social status affected group alignment, line 218:* **[This entire section has been extensively revised and is too long to include here in its entirety.]**

Methods: Conversation content analysis, line 594: **[This entire section has been extensively revised and is too long to include here in its entirety.]**

Discussion, line 300: We did not expect all conversations to be equally successful, and some conversation groups aligned more than others. Specifically, groups with participants perceived as having high social status showed less alignment, while groups with high-centrality participants showed more alignment. This may be because high-status and high-centrality participants behaved differently in conversation.

Discussion, line 304: Participants perceived as high in social status spoke more, gave more orders, and implicitly rejected others' proposals. Despite being rated more influential by their groups, these high-status participants were *less* neurally influential, raising the possibility that their conversation behaviors produced public compliance without private acceptance. This misperception of status cues as markers of

influence may play a pernicious role in the reinforcement of power hierarchies (see e.g., Magee & Galinsky, 2008). By contrast, groups with high-centrality participants had more equal turn-taking, and high-centrality participants did not speak more than others. These participants were both more neurally influential and more likely to be neurally influenced by others—accordingly, their groups achieved greater neural alignment. High-centrality participants may have facilitated this alignment by creating a psychologically safe environment, encouraging others to speak (Edmondson & Lei, 2014), privately accepting and internalizing others' proposals, and rallying their groups around agreeable consensus positions.

Considering the importance of consensus building, and the fact that there were only 9 groups total, I'd like to see a graph showing the consensus building for each of the 9 groups. Descriptively, how much change was there in their opinions and in the ISC score? This seems like an important bit of basic information with which to lead.

We appreciate this concern and have revised the manuscript to clarify the reporting of behavioral and neural alignment across groups and movies. However, we have not added a graph showing alignment on a per-group, per-movie clip basis. This is because we do not want readers to get the wrong idea about our hypothesis: We did not hypothesize that all groups would align behaviorally and neurally for all movies; rather, as noted by the reviewer above, we hypothesized that alignment would depend on features of both the group and their conversation. Instead, we report exploratory analyses of factors affecting alignment, and we note that complete understanding of these factors will require additional research.

Results, line 122: All participants but two reported agreeing with their group's consensus (as rated from -3 to +3 with values > 0 indicating more agreement; $M=1.71$ $t(28)=8.32$, 95% CI=[1.29, 2.13], $p<.001$). Further, each group converged on a different consensus: after conversation, participants' survey answers became more similar to the answers of their conversation group members, compared to those in other groups (within-group participant pairs: $N=61$, mean distance=4.61, 95% CI=[3.9, 5.31]; between-group participant pairs: $N=1210$, mean distance=8.28, 95% CI=[8.11, 8.44]; $t(1269)=-10.08$, $p<.001$) (Supplementary Figure 1). Similarity was measured using city block distance.

Results, line 219: Some groups showed stronger consensus and greater neural alignment than others. Overall, groups that spoke more showed more neural alignment ($b=3.52$, $p=0.007$) and this effect was stronger for groups where conversation turn-taking was more equal among the participants (2-way interaction, $b=-1.34$, $p=0.038$). However, groups with perceived high-status participants (as judged by hypothesis-blind raters) showed less neural alignment as more words were spoken in conversation (2-way interaction, $b=-7.39$, $p=0.021$). By contrast, groups with high-PCA centrality participants showed more neural alignment ($b=1.33$, $p=0.025$).

Results, line 226: Below, we describe similarities and differences between high-centrality and high-status participants in their word choices and general conversation behaviors. [This section continues to describe the new analyses in detail.]

More minor points to consider:

Much of the front end in this paper reminded me of Deutsch and Gerard's (1955 "A Study of Normative and Informational Influences..." Journal of Abnormal and Social Psychology 51(3): 629-36) distinction between normative and informational influence. Informational influence

occurs when someone changes their mind in light of evidence, while normative influence occurs when someone says something to go along with a group, but whether they believe it is unknown. This distinction is relevant when the author state, “[those] offering explanations of social influence have therefore often treated conversation as a means of achieving public compliance, setting internal beliefs aside.” Or “However, it remains unclear whether these conversations shape the way individuals fundamentally see the world, or are merely a means of enforcing public compliance.” This latter quote is directly conceptualizing influence in terms of normative or informational influence. The methods outlined in this paper seem to be a good way to establish whether something is normative or informational, but given the importance of this distinction, I think the authors should address this notion.

We thank the reviewer for noting that public compliance and normative influence are closely related. However, the reported paradigm did not cleanly separate normative and informational influence. When study participants were presented new information, it was typically presented by another participant that professed to believe it, confounding normative and informational factors. We have revised the manuscript to cite Deutsch & Gerard (1955), noting that we could not fully separate normative and informational factors:

Footnote 3: Deutsch & Gerard (1955) draw a related distinction between normative and informational social influence: Normative influence occurs when people conform to a group, whereas informational influence occurs when beliefs are revised in light of new information. Although public compliance implies normative influence, normative and informational factors are confounded in the reported paradigm, preventing separate examination.

With respect to the network centrality results, there is an obvious chicken-and-egg problem that the authors do not address. Are central members central because they are more neurally flexible, or are they neurally flexible because they’re central? I think point is important enough to warrant some discussion.

We agree that this is an interesting question, and have revised the manuscript to raise the issue and cite relevant research:

Discussion, line 316: High-centrality participants flexibly adapted their brain activity to their groups. It may be that this flexibility directly supports conversational skills that facilitate social connection. Previous research on personality and social network centrality points in this direction: People with high self-monitoring personalities (i.e., those who adapt their behavior to the people around them) tend to be more socially central (Fang et al., 2015; Mehra, Kilduff, & Brass, 2001), and they become so by making friends across disconnected cliques (Sasovova, Mehra, Borgatti, & Schippers, 2010). Further, survey measures of self-monitoring, cognitive flexibility, and communication flexibility are closely related and highly correlated (Chesebro & Martin, 2003; Martin & Rubin, 1995). Although we do not know how our participants became central in their social networks, it is plausible that cognitively flexible, consensus-building approaches to conversation enabled them to grow large and diverse groups of friends.

You make the point that brain alignment may shape who befriends whom (page 2). On this topic, you might consider similar work on neural correlates of reciprocity. For example: Zerubavel et al. 2018. PNAS 115(17):4375-4380.

We thank the reviewer for this reference, and have revised the manuscript to cite it:

Background, line 80: Interestingly, Parkinson and colleagues (2018) found that friends in a real-world social network were more likely to have similar brain activity, suggesting brain alignment may play a role in determining who befriends whom, perhaps in conjunction with social reward systems (Zerubavel, Hoffman, Reich, Ochsner, & Bearman, 2018).

The sample size is quite small (by my sociological standards), but I recognize that this varies from field to field (but by psychological standards, why is there only one study?). The actual sample size is only available in the SI. It should be included in the results so that when you say “All but two...” agreed with the consensus, we have a sense of what that means relative to the sample space.

We have revised the manuscript to include relevant sample size information in both the Results and the Methods section (which itself has been moved from the Supplementary Material to the main text):

Experimental paradigm, line 101: To answer these questions, we designed an experiment that allowed us to identify changes in brain alignment that could only be caused by conversation. In Session 1, participants (N=49) watched movie clips with ambiguous narratives during brain scanning using fMRI. Afterward, participants answered a survey assessing their beliefs about each clip’s narrative. In Session 2, participants met in small groups (9 groups; mean group size=4.2) to discuss the movie clips with the goal of reaching a consensus. Group membership was randomly assigned, pursuant to participants’ scheduling constraints. Each group answered the survey presented in Session 1, except expressing the shared view of the group. Participants then rated the influence of the other participants and indicated their personal level of agreement with the consensus. In Session 3, participants re-watched the movie clips during fMRI scanning, along with additional novel clips featuring the same characters. Participants then answered a survey assessing their beliefs about the novel clips. A control group (N=9) skipped Session 2, doing both fMRI sessions without the intervening group conversation.

We understand the reviewer’s concern that the sample size is small and that there is only one study. This is because the reported experimental paradigm is unusually labor intensive, time consuming, and difficult to schedule.

It’s unclear to me why Brokerage was used at all. Centrality also isn’t justified as a measure, but it is a reasonable “default” network measure to explore associations between network measures and neural change. It seems some motivation for these measures is warranted. Why would centrality, or brokerage, or any other network statistic make sense here? More to the point, are there any theories that suggest some relational mechanism that is tapped by a network measure? The reason I bring this up is that the social network component of this comes off as a mere add-on, that you included only because you had the measures. There is very little, if any, “theory” guiding all of this work, and that makes the connection to social networks very tenuous.

We agree with the reviewer that motivating the use of centrality measures is important. Previously, research motivating our choices was mentioned briefly in the description of the conversation analysis. We have revised the manuscript to motivate the use of centrality measures in the introduction:

Background, line 80: Interestingly, Parkinson and colleagues (2018) found that friends in a real-world social network were more likely to have similar brain activity, suggesting brain alignment may play a role in determining who befriends whom, perhaps in conjunction with social reward systems (Zerubavel, Hoffman, Reich, Ochsner, & Bearman, 2018). Previous research has also shown that personality traits (Celli & Polonio, 2013; Liu & Ipe, 2010; Sasovova, Mehra, Borgatti, & Schippers, 2010; Staiano et al., 2012) predict network centrality, suggesting that alignment of brain activity within social networks may depend on the social and conversational behaviors of individuals.

Background, line 88: Here we use ISC to assess whether consensus-building conversation can align thinking within groups. Further, by using a naturalistic experimental paradigm and drawing participants from real-world social networks, we identify relationships between conversation behavior, network centrality, and neural influence.

P. 10 of the SI: you state that you simultaneously modeled “centrality, eigenvector centrality, and brokerage...” Is this a typo or is there another operationalization of centrality aside from eigenvector? Also, does including all three result in collinearity? They should be pretty highly correlated measures.

The reviewer is correct that eigenvector centrality and brokerage are highly correlated. The manuscript describes a "PCA centrality" measure designed to capture variance common to eigenvector centrality and brokerage. PCA centrality was used to account for the collinearity of centrality measures. Because linear regression betas only capture variance unique to each predictor, the beta for PCA centrality captures all of the shared variance, while the betas for eigenvector centrality and brokerage capture only the variance unique to each. We have revised the manuscript to more clearly describe how PCA centrality was calculated, and to specify "PCA centrality" whenever appropriate.

Results, line 186: Participants' centrality in the social network of their school cohort was assessed using brokerage and eigenvector centrality. Brokers are those who connect people who would not otherwise be connected, allowing them to control the flow of information between cliques (Burt, 1992). People with high eigenvector centrality are both well-connected and have many well-connected friends (Bonacich, 1972) (Figure 6). Because brokerage and eigenvector centrality were correlated in our population, we also computed a *PCA centrality* score capturing variation common to both measures (see *Methods: Social network centrality*). P-values reported below were calculated using permutation testing to account for network autocorrelation.

Methods, Social network centrality, line 559: Because eigenvector centrality and brokerage are often highly correlated, principal component analysis was used to create a centrality measure that captured the variance shared by both metrics. PCA was applied using Scikit-learn (Pedregosa et al., 2012), and the social network metrics were projected on the first principal component. The resulting projection is referred to as *PCA centrality*.

Throughout manuscript: [Changed “centrality” to specify “PCA centrality” when appropriate.]

We have also revised the neural influence analysis to be less confusing. We now clarify why PCA centrality, eigenvector centrality, and brokerage are treated separately. We also now control eigenvector centrality and brokerage for each other, so there is no overlap with the PCA centrality analysis.

Results, line 194: Surprisingly, participants who were central in their social networks were more likely to be influenced by others: Alter PCA centrality was correlated with whole-brain influence ($R=0.1$, $p=0.021$, $N=560$). But highly central participants were not more likely to exert influence on others: Ego PCA centrality was not significantly correlated with whole-brain influence ($R=-0.06$, $p=0.145$, $N=560$). Because of hypothesized differences between brokerage and eigenvector centrality (e.g., Burt, 1992, 2010), we assessed the unique contributions of each centrality measure using multiple regression. Predictors included eigenvector centrality and brokerage for both ego and alter, controlling each variable against the others. This regression model modestly predicted whole-brain influence ($F(555,4)=5.02$, $p=0.046$), despite the collinearity of eigenvector centrality and brokerage. Alter eigenvector centrality was associated with more whole-brain influence ($b=335.3$, $p<.001$), while alter brokerage was associated with less whole-brain influence ($b=-270.3$, $p=0.004$). Given that neural influence depends on alter PCA centrality, this result suggests brokers are somewhat less likely to be influenced by others than those with high eigenvector centrality. No ego centrality measures significantly predicted whole-brain influence.

REVIEWER COMMENTS

Reviewer #1 (Remarks to the Author):

I don't really understand the statistics they are using or their justifications.

For example, how they can justify using so many degree of freedom.

Reviewer #2 (Remarks to the Author):

The authors were responsive to the concerns raised in the previous round of review and the manuscript is substantially improved.

Reviewer #3 (Remarks to the Author):

Nice job on the revision. My concerns have largely been addressed. The additional analyses are a nice corrective and illustrate the differential implications of status versus connectivity.

Reviewer #4 (Remarks to the Author):

This is an exciting and creative project and an engaging manuscript, with thoughtful and interesting discussion. My feedback is limited to methodology. I decided to read and make notes on the revised manuscript before I read the previous reviews and rebuttal. Some of what I noticed in reading the revised manuscript ended up overlapping with some of the concerns raised by the original reviewers, meaning that some of these original concerns would benefit from some additional attention.

Before continuing I want to credit the authors for several observations in addition to the engagingness and creativity noted. First, in many respects, these analyses followed best practices (e.g. Chen et al., 2016) and the project includes many recent methodological improvements (multiband acquisitions, hyperalignment) and novel innovations (neural influence). Second, these are quite complicated analyses that are difficult to summarize in a short format; the diagrams and figures are very helpful in this regard. Third, one of the most important and innovative aspects of this project – studying the influence of small group dynamics on the brain – is also a major challenge given the serious limitations of fMRI in small N settings. This is a difficult study to conduct and analyze.

My main concerns pertain to apparent lack of convergence across groups and scans. This is a complicated topic with no easy answers. It is a priori quite reasonable to expect that differences in conversational dynamics would elicit different outcomes in terms of behavioral and neural alignment, and similarly that different video stimuli could elicit different modalities or dimensions of alignment even for the same group. In other words, one would not expect the results to look identical for all groups or for all videos. That being said, though, one would expect some level of convergence a priori (e.g. two groups that landed on similar interpretations should look more similar post-discussion), and given the major noise issues inherent to fMRI, it is common practice to consider some level of convergence (across scans and/or individuals) as relatively more convincing evidence. The way the results are presented makes it difficult to evaluate this for oneself. Most of the main figures collapse results across scans in a way that is understandable but not standard practice, and this leaves open the possibility that only a couple of groups and/or scans could be driving the results. The wording used elsewhere in the manuscript (e.g. "each group aligned in its own way"), scanning Supp Table 5, and the extremely small proportions on the color axes (note: the key Supplemental Figure 2 is missing a color axis entirely) lead me to conclude that this could be the case.

Since the main ISC regression models were run at the level of individual scans, it would be good to see the results at the scan level presented in a way that better permits comparison, in addition to collapsing them across scans. For instance, the main ISC change proportional results (Fig 1) should be presented at the level of individual scans (or scan pairs) in the supplemental information, and the number of groups should be clearly labeled on the color axis. Supp Table 5 and the "cluster reports" (Supp Fig. 6 and onward) do add additional information along these lines, but they do not facilitate comparison. In Supp Figure 6, the legends seem missing and there is not enough information to interpret the figures. (What are the colors in the timeseries? Are the clusters at the scan level across all groups, or reflecting betas for specific groups? What do the MDS results reflect? Note also that MDS methodology seem to be missing from the methods).

Importantly, even though I agree with the authors (in their rebuttal) that their use of an intercept technically serves to control for repeat viewing regardless of group membership, I also wanted to see post-hoc comparisons against the control group, just as Reviewer #1 had requested. Whether conversation matters or not is a central question to the framing, and the main initial ISC models do not test the question as directly as they could. By apparently treating the control group as just another (discussion) group, the models seem to be ignoring the key question of group type (conversation group vs. control group). There are various ways this could be encoded in a future model (with or without an interaction, or collapsing in various ways – see below for one possibility), but this is important to address somehow.

On a related note, it is not clear whether the betas for the control group are or are not included in the proportional brain maps ("proportion of significant movie-group combinations" leaves this ambiguous, as there were indeed significant control group findings, and the methods say "for each group", not "for each experimental group"). This would be important to be explicit about. I doubt that readers would expect the control group to be included here as the results are presented being as "caused by conversation" (Fig 1, similar issue for some other figures as well.).

Pertinent to this issue as well as the "main concerns" above -- perhaps one relatively simple way to present whatever convergence does exist at the scan level would be to add an additional analysis that models pairwise group membership in 3 variables (rather than 9 specific groups): control group, same-discussion-group, other-discussion-group, ignoring group identity.

While recognizing that in this analysis space it can be very difficult to find the right balance between summarizing complex analyses to give a high-level overview of the structure of the data, versus overwhelming and uninterpretable data dumps ... ultimately there is still too much information that is being lost between the actual models run and the very high level aggregated results presented in the main Figures. If what the results end up supporting is ultimately a story about idiosyncratic small-group interpretation-linked brain changes that are largely unshared across groups and across scans, as it seems that it could be, then I'm not sure the sample size is sufficient to be convincing.

Other points: it seems that in many of the correlational analyses, each pair of individuals is treated as an independent data point for each of their scans. There are better ways to do this: a GLMM treating each pair as a random factor, or separate correlation run for each scan, or averaging across all scans for each pair of individuals. The relationships are generally not the strongest as presented, and it is not clear whether they would persist without the increased degrees of freedom due to treating them as independent. (This issue is not addressed by the authors' statement about limitations on generalization.)

It would be good to add R^2 s in the regression models for the neural influence and social network centrality models.

Please define "mass" ("mass of neural change map", "summed positive mass", etc.) as this is a less

conventionally used term.

Please confirm that the social network analyses only consider social network measures within a discussion group and not spanning groups. I believe this is the case based on the statement about the regressions being encoded similarly to the group-level analyses, but I was not sure if the degrees of freedom was or was not consistent with this. It would be good to be explicit about this. (Obviously, if across- group social network measures are included, direct conversation wouldn't be the mechanism.) Similarly, please be explicit about whether neural influence maps consider only within-group pairs or whether across-group comparisons are also included; if the latter, the interpretation would be different.

Response to reviewers

We thank the editor and the reviewers for their kind comments and thorough feedback. We have revised the manuscript to follow the recommendations of Reviewer 4. These revisions include a new fMRI analysis quantifying convergence across groups, as well as new mixed-effects model analyses with random intercepts for participant pairs, accounting for the non-independent structure of the data. We hope that the mixed-effects analyses also address Reviewer 1's concern that the reported degrees of freedom were high.

Additionally, we corrected mislabeled cerebellar activity and made small changes to the manuscript intended to help readers compare the various analyses.

Below, we respond in detail to Reviewers 1 and 4, describing the revisions made to address their comments. Our responses are indented and in blue, with relevant excerpts from the revised manuscript block-quoted and in a smaller typeface. All references are available in the revised manuscript.

Reviewer 1:

I don't really understand the statistics they are using or their justifications.

For example, how they can justify using so many degrees of freedom.

The degrees of freedom were high when participant pairs were the unit of analysis. This is because the best available methods for testing our hypotheses start by calculating alignment between participant pairs. Using participant pairs this way raises two concerns: the paired data are non-independent, and the high degrees of freedom could artificially lower p-values when calculated parametrically. The previously submitted version of the manuscript addressed these issues by calculating p-values using permutation testing for many analyses. The attached revision further addresses these issues by including linear mixed-effects models with random intercepts for participant pairs:

131: Each participant was in many measured pairs, creating non-independent data. Mixed-effects modeling with random intercepts for participant pairs was used to address this non-independence, showing a significant effect of conversation on consensus ($b=-2.78$, 95% CI=[-2.99, -2.56], $p<.001$).

170: To account for the non-independence of participant pairs, a mixed-effects model with random intercepts for participant pairs was used to estimate the effect of meaning change on whole-brain alignment, finding similar results ($b=61.28$, 95% CI=[79.99, 42.58], $p<.001$).

229: To detect neural influence that occurred in different parts of the brain for different pairs of participants, we tested whether network centrality predicted whole-brain neural influence. A mixed-effects model including eigenvector centrality and brokerage for both egos and alters as predictors modestly predicted whole-brain influence (marginal $R^2=0.03$, $p<.001$; see Methods:

Marginal R^2). Alter eigenvector centrality was the only significant predictor, and was associated with more whole-brain influence ($b=335.5$, 95% CI=[92.52, 578.48], $p=0.007$). No ego centrality measures significantly predicted whole-brain influence, and a separate mixed-effects model analysis showed that neither ego nor alter PCA centrality predicted whole-brain influence. To account for the non-independence of participant pairs, these models included random intercepts for participant pairs. Participant pairs were included in these analyses only if they were in the same conversation group.

239: Suggesting that neural influence was related to changes in belief, survey influence calculated over the survey answers of ordered participant pairs was correlated with whole-brain neural influence ($R=0.06$, $p<.001$, $N\text{ pairs}=8620$, p -value calculated using permutation testing), and survey influence predicted neural influence in a mixed-effects model with random intercepts for participant pairs ($b=55.81$, 95% CI=[6.27, 105.35], $p=0.027$).

635: To address the non-independence of participant pairs, linear mixed-effects models with random intercepts for participant pairs were fit the statsmodels package for Python (Seabold & Perktold, 2010).

For only one analysis, mixed-effects modeling yielded a different result: Although we still find a significant effect of alter eigenvector centrality on whole-brain influence, we no longer see an effect of alter PCA centrality (a measure capturing variance shared by both eigenvector centrality and brokerage) on whole-brain influence. This does not change any of the conclusions of the manuscript.

Reviewer 4:

This is an exciting and creative project and an engaging manuscript, with thoughtful and interesting discussion. My feedback is limited to methodology. I decided to read and make notes on the revised manuscript before I read the previous reviews and rebuttal. Some of what I noticed in reading the revised manuscript ended up overlapping with some of the concerns raised by the original reviewers, meaning that some of these original concerns would benefit from some additional attention.

We thank the reviewer for their kind words, generous engagement with our analyses, and constructive, thoughtful suggestions which have made our manuscript clearer in many important ways. Specific point-by-point responses follow.

Before continuing I want to credit the authors for several observations in addition to the engagingness and creativity noted. First, in many respects, these analyses followed best practices (e.g. Chen et al., 2016) and the project includes many recent methodological improvements (multiband acquisitions, hyperalignment) and novel innovations (neural influence). Second, these are quite complicated analyses that are difficult to summarize in a short format; the diagrams and figures are very helpful in this regard. Third, one of the most important and innovative aspects of this project – studying the influence of small group

dynamics on the brain – is also a major challenge given the serious limitations of fMRI in small N settings. This is a difficult study to conduct and analyze.

We thank the reviewer for these very thoughtful and complimentary comments about our work.

My main concerns pertain to apparent lack of convergence across groups and scans. This is a complicated topic with no easy answers. It is a priori quite reasonable to expect that differences in conversational dynamics would elicit different outcomes in terms of behavioral and neural alignment, and similarly that different video stimuli could elicit different modalities or dimensions of alignment even for the same group. In other words, one would not expect the results to look identical for all groups or for all videos.

We agree: We did not expect the results to look identical for all groups and all videos. The submitted revision clarifies this in the introduction:

100: Importantly, we did not hypothesize that simply having a conversation would create neural alignment in a single set of brain areas. Rather, we hypothesized that each group would converge on a different consensus, and therefore would show alignment in different brain areas at different times.

The previous version of the manuscript sometimes used the phrase “caused by conversation.” In all cases, the manuscript has been revised to read “caused by consensus-building conversation.”

That being said, though, one would expect some level of convergence a priori (e.g. two groups that landed on similar interpretations should look more similar post-discussion), and given the major noise issues inherent to fMRI, it is common practice to consider some level of convergence (across scans and/or individuals) as relatively more convincing evidence. The way the results are presented makes it difficult to evaluate this for oneself.

As previously noted by the reviewer, we did not expect the results to look identical for all groups or for all movie clips. This is why we focused on understanding and explaining variation across groups. However, idiosyncratic results must be distinguished from noise. Further, standard practice analysis methods are easier to understand. We have therefore added analyses along the lines suggested by the reviewer in a later comment: We separately estimate the effects of being in a conversation group and being in the control group (no discussion). We also performed an analysis of neural influence including all movies and groups. These analyses directly identify convergence:

135: Increased ISC was observed within conversation groups, supporting the hypothesis that consensus-building conversation can align future brain activity. To capture change in ISC that was convergent across groups, we tested the effect of being in any conversation group (Figure 1 bottom left, Supplementary Figure 2). Importantly, this analysis could not show the effect of

being in a specific group that conversed about a specific movie clip. To address this limitation, we tested the effects of discussing specific movies with specific groups, counting the number of groups with statistically significant results (Supplementary Figure 4). A wider range of brain areas were significant at the movie–group combination level, indicating that the neural effects of conversation depended on who was speaking and what they were speaking about.

215: Unexpectedly, participants who were central in their social networks were more likely to be neurally influenced by others in their conversation groups (Figure 3, Supplementary Figures 7–10, Supplementary Table 5). Multiple regression was used to locate brain areas where neural influence was predicted by the PCA centrality, eigenvector centrality, and brokerage of egos and alters in the same conversation group, across all groups and movie clips (see *Methods: Neural influence* and Supplementary Table 5).

221: Interestingly, ego PCA centrality predicted negative neural influence in right middle temporal gyrus (Supplementary Figure 7), while alter PCA centrality predicted positive neural influence across a range of brain areas (Supplementary Figure 8). Similarly, ego eigenvector centrality predicted negative neural influence in left middle temporal gyrus (Supplementary Figure 9), while alter eigenvector centrality predicted positive neural influence across a range of brain areas (Supplementary Figure 10). These results suggest variance in each centrality measure was associated with different neural processes. Analyses of neural influence specific to each group–movie combination yielded qualitatively similar results (Supplementary Figures 11 and 12, Supplementary Table 6).

534: For the group-specific analyses, predictor matrices were created for each group, with 1s where both participants were in the target group and 0s otherwise. For the across-groups analysis, a single prediction matrix included 1s when both participants were in the same conversation group. For both analyses, an all-1s intercept matrix was used to capture change that occurred whether or not participants were in the same group (i.e., change caused simply by watching a movie clip twice).

607: The effect of social network centrality was assessed using regression analysis over the neural influence maps generated at the level of the participant pair. Predictor matrices were structured similarly to the predictors used in the across-groups change analysis, but instead of 1s, each cell included the social network centrality values of either the ego or the alter. Group-specific predictor matrices zeroed out cells for all but the target group. Because predictor beta estimates made with ordinary least squares regression depend only on the variance unique to that predictor, this approach allowed us to estimate the effects of PCA centrality with one model, and then to estimate the unique effects of eigenvector centrality and brokerage with a second model. Each regression yielded maps of centrality-dependent neural influence.

See Supplementary Table 5 for a summary of analyses that identify convergent activity across movie clips and groups. For visualizations, see Figures 1 and 3, as well as Supplementary Figures 2, 3, 5, and 7–10.

We note that adding these analyses means readers have more to keep track of while reading. Accordingly, we have made revisions indicating whether analyses pertain to convergence across groups or to specific movie–group combinations.

We also agree with the reviewer that participants with similar interpretations should look more similar post-discussion. This is exactly what we found, and the manuscript includes an analysis showing the correlation between similar survey answers and whole-brain alignment. We have revised the manuscript to discuss how this analysis addresses concerns about convergence (and we have added a mixed-effects model analysis as recommended by the reviewer which is detailed later in this response):

169: Similarity of survey answers was correlated with whole-brain alignment, even across groups, including control participants that did not converse ($R=0.09$, $p<.001$, $N\text{ pairs}=3478$). To account for the non-independence of participant pairs, a mixed-effects model with random intercepts for participant pairs was used to estimate the effect of meaning change on whole-brain alignment, finding similar results ($b=61.28$, $95\% \text{ CI}=[79.99, 42.58]$, $p<.001$). These findings suggest that the timings and locations of neural alignment were not idiosyncratic, but were driven by convergence of beliefs (see *Supplementary Material: Comparing behavioral and neural results*).

We also note that the manuscript includes several additional analyses that demonstrate convergence in different ways: We identify spatial overlap in alignment across groups (Supplementary Figures 4, 6, 10, and 11), as well as convergence in the functions associated with aligned brain areas (Figure 2). Further, we identify group-level properties that predict variance in neural alignment (in the section “Network centrality and perceived social status affected group alignment”), showing that the observed variance is systematic.

Most of the main figures collapse results across scans in a way that is understandable but not standard practice, and this leaves open the possibility that only a couple of groups and/or scans could be driving the results. The wording used elsewhere in the manuscript (e.g. “each group aligned in its own way”), scanning Supp Table 5, and the extremely small proportions on the color axes (note: the key Supplemental Figure 2 is missing a color axis entirely) lead me to conclude that this could be the case.

We have revised the main figures (Figures 1 and 3) to show convergent results across all movie clips and groups from the standard practice analysis suggested by the reviewer. (The analysis of novel movie clips seen only during the second scan session depends on a conjunction map of movie- and group-specific analyses so it is now visualized in Supplementary Figures X and X.)

We regret that a color axis label was missing from Supplementary Figure 2. This is corrected in the resubmitted manuscript.

We agree with the reviewer that the reported results are driven by some groups and not others. Importantly, this is consistent with the hypothesis that consensus and neural alignment are causally related: Groups with less consensus should show less alignment.

Critically, the reported group-specific results are statistically significant and survive correction for multiple comparisons.

Since the main ISC regression models were run at the level of individual scans, it would be good to see the results at the scan level presented in a way that better permits comparison, in addition to collapsing them across scans. For instance, the main ISC change proportional results (Fig 1) should be presented at the level of individual scans (or scan pairs) in the supplemental information, and the number of groups should be clearly labeled on the color axis.

As described in our response above, we have added an analysis suggested by the reviewer that collapses across movie clips and groups and facilitates comparison to the control group.

To make the results more readable we have modified the color axes of all conjunction maps to show the number of overlapping group–movie combinations at each voxel rather than a decimal proportion.

Supp Table 5 and the “cluster reports” (Supp Fig. 6 and onward) do add additional information along these lines, but they do not facilitate comparison. In Supp Figure 6, the legends seem missing and there is not enough information to interpret the figures. (What are the colors in the timeseries? Are the clusters at the scan level across all groups, or reflecting betas for specific groups? What do the MDS results reflect? Note also that MDS methodology seem to be missing from the methods).

We have revised the manuscript to clearly identify each element of the cluster reports and to correct ambiguous group and movie clip labeling. We have also added the MDS methodology to the methods section.

Supplementary Material, Cluster reports: Each cluster report below is labeled by movie clip and group and contains the following elements. Upper left: multi-dimensional scaling plot showing relative distances between participants’ patterns of brain activity before conversation (yellow) and after conversation (orange). Bottom left: A spatially contiguous cluster of active voxels that survived multiple comparisons correction for the indicated group and movie clip. Upper middle: Probability of Neurosynth terms given the pattern of brain activity. Right top and middle: Mean BOLD activation over time, with each group member shown in a different color. Right bottom: Change in group-level ISC over time.

631: In *Supplementary Material: Cluster reports* multidimensional scaling (MDS) plots show a 2-dimensional projection of the relative distances between participants BOLD time series, before and after conversation. MDS was performed using Scikit-learn (Pedregosa et al., 2012) using the correlation distance between participants’ BOLD time series.

Importantly, even though I agree with the authors (in their rebuttal) that their use of an intercept technically serves to control for repeat viewing regardless of group membership, I also wanted to see post-hoc comparisons against the control group, just as Reviewer #1 had

requested. Whether conversation matters or not is a central question to the framing, and the main initial ISC models do not test the question as directly as they could. By apparently treating the control group as just another (discussion) group, the models seem to be ignoring the key question of group type (conversation group vs. control group). There are various ways this could be encoded in a future model (with or without an interaction, or collapsing in various ways – see below for one possibility), but this is important to address somehow.

As described in our response above, we have added an analysis suggested by the reviewer that facilitates comparing results that converge across all conversation groups to the control group. Supplementary Figure 2 shows convergent neural alignment in conversation groups, while Supplementary Figure 3 shows that the control group became less neurally aligned.

On a related note, it is not clear whether the betas for the control group are or are not included in the proportional brain maps (“proportion of significant movie-group combinations” leaves this ambiguous, as there were indeed significant control group findings, and the methods say “for each group”, not “for each experimental group”). This would be important to be explicit about. I doubt that readers would expect the control group to be included here as the results are presented being as “caused by conversation” (Fig 1, similar issue for some other figures as well.).

We thank the reviewer for drawing our attention to this issue. When double-checking the analysis code to write this response, we found that in some cases the control group was erroneously included. Because the control group had very few statistically significant clusters, in most cases excluding it yielded slightly improved results. We have also revised this section of the manuscript to explicitly state when the control group was excluded or included:

144: In conversation groups, alignment tended to increase in visual and auditory sensory areas, as well as in higher-order areas associated with the attention and default mode networks, including the temporal parietal junction, angular gyrus, posterior cingulate, medial prefrontal cortex, and temporal pole. These results stand in contrast to the control group (no conversation), where ISC mostly decreased (Supplementary Figure 3). See *Supplementary Material: Cluster reports* and Supplementary Table 5 for all fMRI analysis results.

Figure 1, caption: **Bottom left:** Change in neural alignment caused by consensus-building conversation. Color shows the number of movie–group combinations with significant results at a brain location (including both increases and decreases in alignment; control group excluded). **Bottom right:** Participants whose survey answers became similar showed greater neural alignment (including control group participants that did not converse).

169: Similarity of survey answers was correlated with whole-brain alignment, even across groups, including control participants that did not converse ($R=0.09$, $p<.001$, $N\text{ pairs}=3478$).

Pertinent to this issue as well as the “main concerns” above -- perhaps one relatively simple way to present whatever convergence does exist at the scan level would be to add an

additional analysis that models pairwise group membership in 3 variables (rather than 9 specific groups): control group, same-discussion-group, other-discussion-group, ignoring group identity.

We thank the reviewer for this excellent suggestion and have added an analysis along these lines to the manuscript, as described in our response above.

While recognizing that in this analysis space it can be very difficult to find the right balance between summarizing complex analyses to give a high-level overview of the structure of the data, versus overwhelming and uninterpretable data dumps ... ultimately there is still too much information that is being lost between the actual models run and the very high level aggregated results presented in the main Figures. If what the results end up supporting is ultimately a story about idiosyncratic small-group interpretation-linked brain changes that are largely unshared across groups and across scans, as it seems that it could be, then I'm not sure the sample size is sufficient to be convincing.

We hope that the changes and additional analyses described above help link the low- and high-level aspects of the study, addressing the reviewer's concerns. We note that although the group sizes are small, the reported group-specific results are statistically significant and survive correction for multiple comparisons.

Other points: it seems that in many of the correlational analyses, each pair of individuals is treated as an independent data point for each of their scans. There are better ways to do this: a GLMM treating each pair as a random factor, or separate correlation run for each scan, or averaging across all scans for each pair of individuals. The relationships are generally not the strongest as presented, and it is not clear whether they would persist without the increased degrees of freedom due to treating them as independent. (This issue is not addressed by the authors' statement about limitations on generalization.)

To address the concern that the reported results may have been affected by the non-independence of participant pairs, we conducted mixed effects model analyses which found largely the same results. The manuscript now reports mixed effects model betas in addition to correlation coefficients:

131: Each participant was in many measured pairs, creating non-independent data. Mixed-effects modeling with random intercepts for participant pairs was used to address this non-independence, showing a significant effect of conversation on consensus ($b=-2.78$, 95% CI=[-2.99, -2.56], $p<.001$).

170: To account for the non-independence of participant pairs, a mixed-effects model with random intercepts for participant pairs was used to estimate the effect of meaning change on whole-brain alignment, finding similar results ($b=61.28$, 95% CI=[79.99, 42.58], $p<.001$).

229: To detect neural influence that occurred in different parts of the brain for different pairs of participants, we tested whether network centrality predicted whole-brain neural influence. A mixed-effects model including eigenvector centrality and brokerage for both egos and alters as

predictors modestly predicted whole-brain influence (marginal $R^2=0.03$, $p<.001$; see Methods: Marginal R^2). Alter eigenvector centrality was the only significant predictor, and was associated with more whole-brain influence ($b=335.5$, 95% CI=[92.52, 578.48], $p=0.007$). No ego centrality measures significantly predicted whole-brain influence, and a separate mixed-effects model analysis showed that neither ego nor alter PCA centrality predicted whole-brain influence. To account for the non-independence of participant pairs, these models included random intercepts for participant pairs. Participant pairs were included in these analyses only if they were in the same conversation group.

239: Suggesting that neural influence was related to changes in belief, survey influence calculated over the survey answers of ordered participant pairs was correlated with whole-brain neural influence ($R=0.06$, $p<.001$, $N\text{ pairs}=8620$, p -value calculated using permutation testing), and survey influence predicted neural influence in a mixed-effects model with random intercepts for participant pairs ($b=55.81$, 95% CI=[6.27, 105.35], $p=0.027$).

635: To address the non-independence of participant pairs, linear mixed-effects models with random intercepts for participant pairs were fit the statsmodels package for Python (Seabold & Perktold, 2010).

For only one analysis, mixed-effects modeling yielded a different result: Although we still find a significant effect of alter eigenvector centrality on whole-brain influence, we no longer see an effect of alter PCA centrality on whole-brain influence. This does not change any of the conclusions of the manuscript. Because this result is no longer easy to explain in terms of simple correlation, we have removed the corresponding correlation visualization from Figure 3.

We note that, where appropriate, p -values for correlation coefficients were calculated using permutation testing, which partially accounted for non-independence between participant pairs. That said, we agree that the manuscript is improved by the inclusion of mixed effects models, as it is good to show that the results hold when using different methods.

It would be good to add R^2 s in the regression models for the neural influence and social network centrality models.

We note that R -squared is not well-defined for mixed-effects models. As recommended by Nakagawa & Schielzeth (2013), we now report a marginal R -squared:

230: A mixed-effects model including eigenvector centrality and brokerage for both egos and alters as predictors modestly predicted whole-brain influence (marginal $R^2=0.03$, $p<.001$; see Methods: Marginal R^2).

617: Marginal R^2 captures the variance accounted for by fixed-effects only and was calculated using the procedure of (Nakagawa & Schielzeth, 2013). The marginal R^2 p -value was calculated using permutation testing: The analysis was repeated with 2000 different permutations of the DV to create a null distribution of marginal R^2 values, and the p -value was the fraction of the null values as or more extreme than the empirical marginal R^2 value.

Please define “mass” (“mass of neural change map”, “summed positive mass”, etc.) as this is a less conventionally used term.

Throughout the revised manuscript we have replaced “mass” with “sum” or “mean” as appropriate.

Please confirm that the social network analyses only consider social network measures within a discussion group and not spanning groups. I believe this is the case based on the statement about the regressions being encoded similarly to the group-level analyses, but I was not sure if the degrees of freedom was or was not consistent with this. It would be good to be explicit about this. (Obviously, if across- group social network measures are included, direct conversation wouldn’t be the mechanism.) Similarly, please be explicit about whether neural influence maps consider only within-group pairs or whether across-group comparisons are also included; if the latter, the interpretation would be different.

We thank the reviewer for pointing this out, and we have revised the manuscript to clarify that participant pairs were included in this analysis only if they were in the same conversation group:

217: Multiple regression was used to locate brain areas where neural influence was predicted by the PCA centrality, eigenvector centrality, and brokerage of egos and alters in the same conversation group, across all groups and movie clips (see *Methods: Neural influence* and Supplementary Table 5).

REVIEWER COMMENTS

Reviewer #1 (Remarks to the Author):

My concern about the statistics has now been throughly and sucessfully addressed.

Reviewer #2 (Remarks to the Author):

The authors were responsive to the issues raised by Reviewer 4 and the manuscript has been further improved.

Reviewer #3 (Remarks to the Author):

I have reviewed this paper previously - in the first round I believe. The authors have done an excellent job of responding to reviewer concerns. At this stage, I have two nitpicky comments.

First, in the third sentence, what's with the use of "klatches." Is someone trying to win a bet by using a random word in publication?

Second, line 94, "this is the first study to..." Sentences like this make me nervous. I'm not certain of this, and I don't think the authors should be either.

Reviewer #4 (Remarks to the Author):

The authors have addressed many of my concerns. There is still some ambiguous or missing information that needs to be clarified though.

* I'm not sure what "movie-group combinations" actually means, and importantly for interpreting the results, how many of them there are meant to be. There are various points in the text where I thought the authors meant 45 (5 movies x 9 groups), 5 (movies), and 9 (groups). This term needs to be defined explicitly, and the text needs to be carefully checked to avoid confusion and make sure the authors are using it consistently. Importantly, any figure that includes a count statistic needs to clearly label what the denominator is, so readers can interpret in context. For e.g. Supp. Fig. 4, the color axis max of 5 is incredible convergence if it's 5/5, moderate if it's 5/9, or small if 5/45. (Granted, overwhelming convergence isn't the point here -- but readers need to know what they are looking at.)

* There is a lack of information in the methods and many figure captions about how the authors go from before-after scan pairs - the level described in the methods - to the single brain maps which seem to aggregate across 5 different scans/movies. This needs to be made explicit in the methods as well as in figure captions. (This also relates to the previous comment for the count-based figures.) For the new mixed LMs, I presume those models included multiple movie clips in a single model, but this needs to be made explicit as well. And importantly, how one goes from five betas from five scan-pair regressions to a single beta on a brain map for the initial results and for the social network analyses as well (both the brain map analyses and the collapsed whole-brain alignment analyses).

* Related to above -- for quite a few figures, I really wasn't sure what I was looking at - it would also be helpful to be more explicit about which analyses correspond with which figures. (e.g. Is Fig 3 a conventional regression somehow aggregated, or is it one of the mixed LMs?) Many of the figure captions are worded in a way to suggest they could be plotting either raw ISC differences ("ISC change") or betas from the regression models - I suspect the latter, but it needs to be clear and labeled.

* For Figure 1 lower left, the figure caption and the colorbar labels are no longer in sync. It looks like

one was updated for the revision but not the other, and it's not clear what the figure is visualizing at this point.

I appreciated the authors' text regarding the limits of generalization based on both the statistical and sampling approaches. I also enjoyed reading the very engaging, interesting, and well-written discussion. As the authors acknowledge -- these results are not large effects and could be considered largely descriptive. In light of that, I think it would be appropriate to both tone down the discussion in places and be more explicit about this, to ensure readers do not miss this important caveat.

Other smaller points:

- * Methods needs to describe video lengths and # TRs per scan.

- * For Fig. 3 would be helpful to unpack in the caption the interpretation of the color axis (e.g., red means what, more concretely).

Response to reviewers

We thank the editor and the reviewers for their time and effort. We have revised the manuscript to follow the recommendations of Reviewers 3 and 4.

Below, we respond in detail to Reviewers 3 and 4, describing the revisions made to address their comments. Our responses are indented and in blue, with line-numbered excerpts from the revised manuscript block-quoted and in a smaller typeface. All references are available in the revised manuscript.

Reviewer 3:

I have reviewed this paper previously - in the first round I believe. The authors have done an excellent job of responding to reviewer concerns.

We thank the reviewer for their kind comment.

At this stage, I have two nitpicky comments. First, in the third sentence, what's with the use of "klatches." Is someone trying to win a bet by using a random word in publication?

In the United States, coffee klatsches are symbolically important because of their role in casual political discussions that shape voting behavior. That said, we do not want to be too U.S.-centric, so the revised manuscript uses "coffee shops."

Second, line 94, "this is the first study to..." Sentences like this make me nervous. I'm not certain of this, and I don't think the authors should be either.

We have removed this and similar claims. The revision now reads:

93: This study evaluates how face-to-face, open-ended social interactions shape participants' future neural responses to complex, naturalistic stimuli.

320: The present study shows that consensus-building conversation strengthens the neural alignment of group members across a wide range of brain areas.

Reviewer 4:

The authors have addressed many of my concerns. There is still some ambiguous or missing information that needs to be clarified though.

We deeply appreciate the reviewer's helpful, thorough feedback and the considerable time they have spent with our work.

I'm not sure what "movie-group combinations" actually means, and importantly for

interpreting the results, how many of them there are meant to be. There are various points in the text where I thought the authors meant 45 (5 movies x 9 groups), 5 (movies), and 9 (groups). This term needs to be defined explicitly, and the text needs to be carefully checked to avoid confusion and make sure the authors are using it consistently. Importantly, any figure that includes a count statistic needs to clearly label what the denominator is, so readers can interpret in context. For e.g. Supp. Fig. 4, the color axis max of 5 is incredible convergence if it's 5/5, moderate if it's 5/9, or small if 5/45. (Granted, overwhelming convergence isn't the point here -- but readers need to know what they are looking at.)

We have revised the manuscript to clarify the meaning of “movie–group combination” and to state the total:

140: We use the term "movie–group combination" to refer to effects that were assessed for a specific group watching a specific movie; 5 movies by 9 groups gives 45 possible movie–group combinations.

Supplementary Figure 4, caption: The maximum possible number of overlapping clusters was 45, and the maximum observed number of overlapping clusters was 5 (or 11%).

Supplementary Figure 6, caption: The maximum possible number of overlapping clusters was 45, and the maximum observed number of overlapping clusters was 8 (or 18%).

Supplementary Figure 11, caption: The maximum possible number of overlapping clusters was 45, and the maximum observed number of overlapping clusters was 3 (or 7%).

Supplementary Figure 12, caption: The maximum possible number of overlapping clusters was 45, and the maximum observed number of overlapping clusters was 3 (or 7%).

We have revised the color bars on the relevant supplementary figures to report the proportion of movie–group combinations as a fraction.

We have confirmed that “movie–group” combination is used consistently throughout the manuscript.

There is a lack of information in the methods and many figure captions about how the authors go from before-after scan pairs – the level described in the methods – to the single brain maps which seem to aggregate across 5 different scans/movies. This needs to be made explicit in the methods as well as in figure captions. (This also relates to the previous comment for the count-based figures.) For the new mixed LMs, I presume those models included multiple movie clips in a single model, but this needs to be made explicit as well. And importantly, how one goes from five betas from five scan-pair regressions to a single beta on a brain map for the initial results and for the social network analyses as well (both the brain map analyses and the collapsed whole-brain alignment analyses).

We thank the reviewer for catching this issue. The resubmitted manuscript clarifies:

542: For analyses including all movie clips, the relevant change and predictor matrices were concatenated, supporting the analysis of multiple clips using a single model.

616: For analyses including all movie clips, the relevant influence and predictor matrices were concatenated, supporting the analysis of multiple clips using a single model.

Figure 1, caption: Method and change in alignment across all movie clips and groups. Brain maps show beta weights from a single regression model incorporating all movie clips and conversation groups (see *Methods: Conversation-induced change in ISC*).

Figure 3, caption: Neural influence across all movie clips and groups. [...] All movie clips and groups were analyzed using a single regression model (see *Methods: Neural influence*).

Related to above -- for quite a few figures, I really wasn't sure what I was looking at -- it would also be helpful to be more explicit about which analyses correspond with which figures. (e.g. Is Fig 3 a conventional regression somehow aggregated, or is it one of the mixed LMs?) Many of the figure captions are worded in a way to suggest they could be plotting either raw ISC differences ("ISC change") or betas from the regression models -- I suspect the latter, but it needs to be clear and labeled.

We have revised the figure captions to make it completely clear that they plot betas from regression models, and to reference the relevant Methods subsections:

Figure 1, caption: Linear regression was used to model change in inter-subject correlation (ISC) (see *Methods: Conversation-induced change in ISC*). **Bottom left:** Change in neural alignment caused by consensus-building conversation. Color shows the beta weight for being in any conversation group (control group excluded), across all movie clips.

Figure 3, caption: Neural influence across all movie clips and groups. [...] All movie clips and groups were analyzed using a single regression model (see *Methods: Neural influence*). Brain maps show the beta weights for ego PCA centrality (middle) and alter PCA centrality (bottom) (see *Methods: Social network centrality*).

All supplementary figure captions were revised in the same way.

For Figure 1 lower left, the figure caption and the colorbar labels are no longer in sync. It looks like one was updated for the revision but not the other, and it's not clear what the figure is visualizing at this point.

We thank the reviewer for pointing this out. We have corrected the figure caption as follows:

Figure 1, caption: Color shows the beta weight for being in any conversation group (control group excluded), across all movie clips.

I appreciated the authors' text regarding the limits of generalization based on both the statistical and sampling approaches. I also enjoyed reading the very engaging, interesting, and well-written discussion. As the authors acknowledge -- these results are not large effects and could be considered largely descriptive. In light of that, I think it would be appropriate to both tone down the discussion in places and be more explicit about this, to ensure readers do not miss this important caveat.

We have revised the discussion to emphasize the limited scope of generalization, and have toned down some statements where readers might have been tempted to over-generalize:

360: Although the present results are suggestive, they are limited in scope. The reported effects are relatively small, and the statistical methods used do not guarantee generalization to the wider population.

381: However, the present results show that consensus-building conversation can align neural responses within groups, and that this alignment can generalize to novel stimuli that were not discussed.

389: Additionally, we would not have observed neural alignment if the participants tended to represent the same concepts using different neural processes.

392: Finally, if neural alignment is a core part of consensus-building conversation, then it poses a challenge to the claim that language did not evolve for communication, but instead for organizing individual thought.

398: The tight, ever-evolving coupling between your thoughts and my thoughts, corresponding to tight neural alignment across our brains, is a plausible mechanism for building group realities based on shared language.

Other smaller points:

* Methods needs to describe video lengths and # TRs per scan.

We have revised Supplementary Table 4 to include the length of the videos in seconds. The Methods section refers readers to this table:

448: See Supplementary Table 4 for titles, presentation order, and duration in seconds and TRs.

* For Fig. 3 would be helpful to unpack in the caption the interpretation of the color axis (e.g., red means what, more concretely).

We thank the author for the push to make this caption clearer. The revision concisely states how the finding relates to the colored regions of the brain maps:

Figure 3, caption: Neural influence reflects the movement of the alter's BOLD time series toward the ego's initial BOLD time series. When egos had higher PCA centrality, alters moved away from them after conversation (top brain map, blue areas). When alters had higher PCA centrality, they moved toward their egos after conversation (bottom brain map, red areas).

REVIEWERS' COMMENTS

Reviewer #4 (Remarks to the Author):

The authors have addressed my feedback. No further comments...